# Memory for rewards guides retrieval
Juliane Nagel [1,2,3] ✉, David Philip Morgan [1,2,3], Necati Çağatay Gürsoy [1,2,3], Samuel Sander[1,2,3], Simon Kern [1,2,3] & Gordon Benedikt Feld[1,2,3,4] ✉

Rewards paid out for successful retrieval motivate the formation of long-term memory. However, it has been argued that the Motivated Learning Task does not measure reward effects on memory strength but decision-making during retrieval. We report three large-scale online experiments in healthy participants ($N = 200$, $N = 205$, $N = 187$) that inform this debate. In experiment 1, we found that explicit stimulus-reward associations formed during encoding influence response strategies at retrieval. In experiment 2, reward affected memory strength and decision-making strategies. In experiment 3, reward affected decision-making strategies only. These data support a theoretical framework that assumes that promised rewards not only increase memory strength, but additionally lead to the formation of stimulus-reward associations that influence decisions at retrieval.

Rewards can enhance learning and memory (for a review, see Miendlar-zewska et al.[1]). This occurs when they are experienced as a consequence of an action (operant conditioning), but even the anticipation of a reward at retrieval is sufficient to motivate encoding[2–6]. In this framework, items associated with higher rewards are retained better, usually assessed with a version of the "Motivated Learning Task"[7–9]. In this paradigm, participants learn stimuli cued with varying amounts of reward, which is paid out for later recognizing the studied stimuli (targets) among new ones (lures).

Dopamine plays a key role in prioritizing memories[10], presumably via enhanced encoding[11] and increased consolidation[6]. It determines which information enters long-term memory via the hippocampal-VTA loop[12]. Presenting reinforcers during learning elicits a dopaminergic response[13], which modulates neuroplasticity[14] and thereby increases the memory strength of the studied items[15,16]. Additionally, animal models suggest that dopamine tags memory traces for preferential reactivation during sleep[17,18].

In the Motivated Learning Task, researchers typically rely on the hit rate (number of recognized targets) to investigate the impact of reward on memory[2,7]. However, without correction for false alarms, (i.e., incorrectly identifying a lure as target), the hit rate cannot be used to disentangle memory strength (discriminability, the ability to discriminate between old and new stimuli[19]) from decision-making strategies (response bias, the tendency to respond "old" during recognition[19]). Memory strength and decision-making strategies can be distinguished by considering the hit rate and the false alarm rate together (calculating discriminability $d'$ and response bias $C$). However, in the standard implementation of the Motivated Learning Task lures are not assigned a reward, which makes it impossible to calculate those indices per reward level.

In three thoughtful experiments, Bowen et al.[20] delineated the influence of reward on memory strength and decision-making strategies. In their modified version of the Motivated Learning Task, rewards were associated with a category (indoor vs. outdoor scenes) rather than with individual stimuli, i.e., lures had an implied reward level. In experiment 1, Bowen et al.[20] found no effect of reward on memory strength, but participants were more inclined to respond "old" to high reward items. In experiment 2 and 3, the researchers eliminated the influence of reward on decision-making strategies by penalizing false alarms contingent on their reward category. Similarly, Marini et al.[3] presented reward information during study and test, and show that reward information can bias decisions during retrieval, although the study sample was small.

Although these studies suggest that the effects of reward on hit rate can be explained by a shift in decision strategies, they supplied the participants with perfect information about the reward contingencies at retrieval. To identify the crucial involvement of stimulus-reward associations formed during learning for decision-making at retrieval and independent effects of rewards on memory strength, we conducted a series of three large-scale online experiments.

## Methods
### Design
Generally, across all three experiments, methods and materials were highly similar, and any deviations from the general methods between the three experiments are detailed in the relevant sections. Participants underwent an adapted version of the Motivated Learning Task (as e.g. reported in Adcock et al.[2]) described below (also see Fig. 1). All tasks were programmed using JsPsych, a collection of JavaScript plugins for browser-based experiments[21].

[1]Clinical Psychology, Central Institute of Mental Health, Medical Faculty Mannheim, University of Heidelberg, Mannheim, Germany. [2]Addiction Behavior and Addiction Medicine, Central Institute of Mental Health, Medical Faculty Mannheim, University of Heidelberg, Mannheim, Germany. [3]Psychiatry and Psychotherapy, Central Institute of Mental Health, Medical Faculty Mannheim, University of Heidelberg, Mannheim, Germany. [4]Department of Psychology, University of Heidelberg, Heidelberg, Germany. ✉e-mail: juliane.nagel@zi-mannheim.de; gordon.feld@zi-mannheim.de

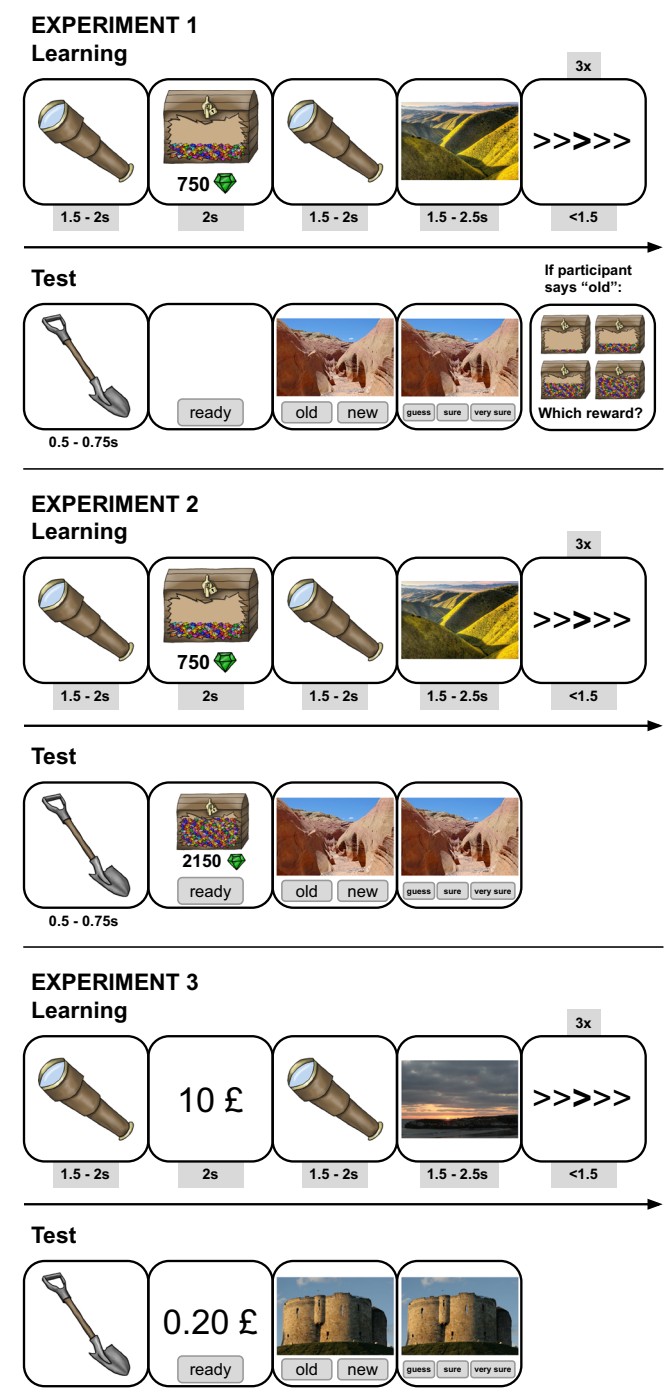

**EXPERIMENT 1**
**Learning**

**Test**

**EXPERIMENT 2**
**Learning**

**Test**

**EXPERIMENT 3**
**Learning**

**Test**

**Fig. 1 | Motivated Learning Task.** Example trials of the Motivated Learning Task for the learning and test phases of experiment 1, 2 and 3. During the learning phase, participants had to memorize 128 images. Each image was associated with a reward presented before the image (gems in a treasure chest in experiments 1 and 2, or cash (GBP), displayed as pictures of a bank note/coin in experiment 3). Trials of the Motivated Learning Task were separated by three trials of a flanker task to prevent rehearsal. During the test phase, participants had to determine whether the image they were presented with was "old" (i.e., shown during the learning phase) or "new" (i.e., not shown during the learning phase) and indicate their confidence in their choice on a 3-point likert scale (i.e., "guess", "sure", "very sure"). In experiment 1, if participants said that an image is "old" they were asked to indicate which reward they thought the image was associated with. In experiments 2 and 3, the reward was shown before the picture also in each trial of the test phase. In experiment 2, the reward was either congruent or incongruent to the reward shown to the participant during the study phase. In experiment 3, rewards shown during the test phase were always congruent to the reward shown during the study phase. In all experiments, participants who made a hit were rewarded the amount associated with the image. For a correct rejection, they received the mean value of all possible rewards and if they made an incorrect decision, they lost the mean value of all possible rewards. The example stimuli shown for experiments 1 and 2 have been uploaded under the Public Domain Mark 1.0 Universal license by Bureau of Land Management California and Jeff Hollet on Flickr. The example stimuli shown for experiment 3 are photographs taken by Gordon Feld. The graphics used as fixation markers and reward displays were drawn by the Juliane Nagel.

This study's design and its analysis were preregistered on the OSF (Experiment 1, April 26, 2021, https://osf.io/ufwmv; Experiment 2, August 4, 2021, https://osf.io/fhzqs; Experiment 3, February 28, 2022, https://osf.io/nefud).

**Participants**

Participants across all three experiments were recruited via Prolific (https://www.prolific.co/), an online work-sourcing site. The number of participants and demographic information for each experiment are presented in Table 1. In order to be included in the experiment and/or data analysis, participants had to meet the inclusion and exclusion criteria presented in Supplementary Tables 1 and 2 (see Supplementary Tables 3–5 for information about how many participants were excluded for which reasons). Participants who did not meet the exclusion criteria were re-sampled until the desired sample sizes were achieved. Oversampling was permitted to facilitate the data collection process. All participants provided informed consent prior to participation, and all experiments were approved by the ethics committee of the University Medical Faculty Mannheim, University of Heidelberg. For experiment 1 and experiment 2, participants received £ 6.50 for completing both parts of the study. At the end of the experiment, participants were ranked according to their score (gems won in the task) and received a bonus payment (rank 1-4: £ 25; 5-8: £ 20; 9-12: £ 15; 13-50: £ 7.50). In experiment 3, participants received £7.60 for completing both parts of the study. Additionally, they earned a bonus based on their performance on the task: They received the average reward they won across all trials of the test phase, multiplied by 2, i.e., participants could additionally win between £ 0 and £ 10.20.

**Procedure**

All participants completed each experiment in a web browser using their computer or laptop. The first part of the study began with a set of demographic questions (age, gender, highest level of education, native language). They then completed the Stanford Sleepiness Scale[22] (SSS) and the Psychomotor Vigilance Task[23] (PVT). Afterwards, participants were provided with instructions for the Motivated Learning Task, which included a description of both the learning and the test phase, and the reward and punishment contingencies. Participants practiced both the learning and test phase of the Motivated Learning task (3 targets, 3 lures), and then answered a set of questions to ensure they understood the reward and punishment contingencies of the task. The first part of the study ended with the learning phase of the Motivated Learning Task. Participants returned for the second

**Table 1 | Demographics for each of the three experiments**

| | experiment 1 | experiment 2 | | experiment 3 |
| --- | --- | --- | --- | --- |
| | | congruent | incongruent | |
| n | 200 | 103 | 102 | 187 |
| sex (female/male) | 106/94 | 54/49 | 48/54 | 103/84 |
| gender (women/men/non-binary/undisclosed) | 104/94/2/0 | 54/49/0/0 | 46/54/1/1 | 100/84/3/0 |
| age M(SD) | 24.34 (3.56) | 22.91 (3.65) | 22.89 (3.62) | 26.91 (4.90) |
| mother tongue English | 73.00% | 87.38% | 90.20% | 85.03% |

part of the study ~22–26 h later. Due to a coding error, the retention interval varied slightly for some participants (for each experiment, 4.00%, 2.94%, 0.54% of participants deviated by >60 min from the planned retention interval), but for each participant, the two sessions were separated by at least one night of sleep and the results were not affected by excluding these participants. When participants returned for the second session, they completed the SSS and the PVT again, and had the opportunity to read the instructions for the test phase of the Motivated Learning Task again. Once again, they had to pass the questions about the reward contingencies. Participants then completed the test phase of the Motivated Learning Task, and were debriefed afterwards.

### Stimuli

For experiment 1 and 2, we collected landscape images from the Creative Commons online repository and assessed them in terms of aesthetics, composition, familiarity, memorability, and whether or not the participants knew the images in a pilot study. Participants recruited via Prolific ($N = 152$) provided ratings on these aspects (each participant rated 73 pictures out of a stimulus pool of 975 pictures, resulting in $M = 11.38$ ratings per picture) and additionally completed a subsequent recognition memory test to collect a measure of memory accuracy for each picture. Based on the measures collected in the pilot study, we divided the images into small subsets (to achieve the required number of final stimuli, we removed the stimuli that received the highest "known" ratings) with properties that were as similar as possible using the R package *anticlust*[24]. The sets were then balanced across task variables (reward level, exposure duration, image category (target/lure) and in experiment 2, experimental group). This way, each participant was presented with a unique stimulus set in the Motivated Learning Task, but variance in stimulus properties across sets was minimized. In experiment 3, we used the stimulus set previously used in Feld et al.[8] and Alizadeh Asfestani et al.[25], consisting of indoor scenes, outdoor scenes and buildings.

### Motivated Learning Task

Because memory tasks with a lot of trials can be quite exhausting for participants, we wanted to increase motivation and reduce attrition rates using a gamified version of the Motivated Learning Task. We embedded the task in a cover story, where participants were recruited as crew members on a pirate ship. As part of their duties on the pirate ship, their goal was to scout for treasure at different locations (corresponding to the learning phase). The treasures were chests containing varying amounts of gems (i.e., different reward magnitudes). Participants navigated between the locations via the Flankers Task, which was stylized as arrows on a treasure map. When participants returned the next day, they went out to retrieve the treasures with their crewmates (corresponding to the test phase), visiting both old locations (targets) and new locations (lures). When participants correctly recognized an old location, they could collect the treasure (i.e., they received the reward associated with the target). If they missed an old location, the crew's captain punished them by taking gems away from them. Other pirate crews inhabited new locations, so if participants mistakenly identified a new location as old, the rival pirate crew would steal some of their gems. This is why the crew's captain rewarded pirates who prevent such a situation, i.e., participants received gems if they correctly identified a new location as new.

Our cover story is highly similar to the interrogative cover story described by Sinclair et al.[26], who found that participants who received interrogative instructions (as compared to imperative instructions) performed better in a recognition memory task, had better memory for rewards associated with the studied items, and demonstrated greater effects of memory-enhancing reward effects.

The Motivated Learning Task was split into two phases, the learning phase and the test phase. In the learning phase, participants were asked to memorize 128 landscape images for a later memory test. Four additional images at the beginning and at the end of the task, which did not appear in the later memory test, buffered primacy and recency effects. Each trial started with a fixation (a spyglass; 1500–2000 ms), followed by a reward (2000 ms). There were different reward amounts (see description of each

experiment for specific amounts) which corresponded to the amount participants could win if they correctly identified the subsequent image as old during the recognition test 24 h later. After a second fixation (1000–1500 ms), a landscape image was presented, for one of four possible exposure durations (see description of each experiment) to control for encoding strength effects unrelated to reward. This was done as a positive control of our paradigm as well as to prepare future studies that will intervene during the retention period, because exposure duration has previously been found to increase hit rate, without interacting with reward (Feld et al.)[8]. After each image, participants completed three trials of a flanker task, where they were shown a series of arrows and had to press the arrow key that corresponded to the direction of the middle arrow highlighted in bold. Trials in this task were either congruent (e.g., >>>>>) or incongruent (e.g., >><>>). Halfway through the learning phase, participants were offered a short break lasting a maximum of 2 min and a minimum of 30 s.

In the test phase, participants were shown the landscape images they had learned previously (i.e., targets; 128 images) and new images (i.e., lures; 128 images). In each trial, participants were shown a fixation (a shovel; 500–750 ms) followed by a "ready" button which participants clicked before seeing the image. This ensured that participants' mouses were in approximately the same position before seeing the image, to make reaction times comparable across trials. Whilst the image was shown participants had to decide whether the image was "old" (i.e., they did see the image during the learning phase) or "new" (i.e., they did not see the image during the learning phase). Identifying a target as old was a hit and participants won the reward associated with the image, and identifying a lure as new (correct rejection) was rewarded with the average of all possible rewards. Making a mistake (false alarm: identifying a new image as old; miss: identifying an old image as new) resulted in the loss of the average reward. Afterwards, participants were asked to rate their confidence in their old/new decision on a 3-point scale ("guess", "sure", "very sure"). See Fig. 1 for a description of the task for all three experiments.

In the first experiment ($N = 200$), we attempted to disentangle memory strength and decision-making strategies by adjusting the paradigm in such a way that the false alarm rate per reward level could be calculated without using categories or showing the reward during recognition testing. Participants learned several pictures associated with different amounts of points (gems), which were earned for correctly identifying the pictures in a recognition memory test 24 later. After identifying a picture as either old or new, we asked participants for their confidence in their decision ("guess", "sure", "very sure"). Previous studies using the Motivated Learning Task have found that high confidence responses are sensitive to reward, but found no evidence for (or against) a reward effect in low confidence items[2]. Whenever a participant identified an image as old during recognition, we asked them how many gems they expected to win for their answer - irrespective of whether the image was in fact a target or a lure (see Fig. 1). This gave us information about which reward participants expected to receive whenever they made a false alarm. Note that this experiment also had other pre-registered goals, and the associated analyses can be found in Supplementary Note 4. Disentangling memory strength and decision-making strategies was not an explicit goal of the preregistration for experiment 1, because we assumed our modification to the paradigm would enable us to calculate a false alarm rate per reward level based on our previous studies[8,25]. This turned out to be wrong, which led to experiments 2 and 3. Descriptive performance measures for the different tasks and control tasks can be found in Supplementary Table 6. Reward levels were 50, 750, 1450 and 2150 gems, and exposure durations were 1500, 1833, 2167 and 2500 ms.

In experiment 2 ($N = 205$), we explicitly assigned a reward to lures by presenting the reward information not only during the learning phase, but also during the test phase (see Fig. 1). Participants were informed that the reward presented before each image during the learning and the test phase corresponds to the amount of reward that is paid out according to the same contingencies as in experiment 1. We did not mention whether the reward presented during the test phase matched the reward presented during the learning phase. However, reward congruency between learning phase and

test phase was manipulated between two groups: In the congruent group ($n = 103$), the reward presented for targets during the test phase (shown reward) always corresponded to the reward presented during the learning phase (true reward). In the incongruent group ($n = 102$), the reward presented for targets was completely orthogonal to the true reward the picture was associated with during the learning phase (i.e., in 25% of target trials, true and shown reward matched). In both groups, lures were presented with a pseudo-random reward level (each reward level appeared equally often across all lures). As in experiment 1, we asked participants for their confidence in their old/new decision ("guess", "sure", "very sure"), but since rewards were presented during the test phase, we did not ask about their reward expectations. We expected reward to exert its influence on memory because it strengthens encoding during and consolidation after learning, not because it changes decision-making strategies during test. That is, in both the congruent and incongruent group, we preregistered to find an effect of true reward on hit rate. In the congruent group, we expected to find an effect of reward on memory strength ($d'$). In the incongruent group, we expected to find no effect of shown reward on memory strength, but hypothesized the shown reward to affect criterion. We also preregistered an effect of true reward on memory strength and possibly criterion in the incongruent group, but later realized that this part of the preregistered model could not be calculated as planned (for details, see Supplementary Note 5). We further predicted that the previously observed reward effect in the Motivated Learning Task mainly reflects improved memory strength, i.e., we did not expect to find an effect of true reward in decision-making strategies (criterion). However, we acknowledged in our preregistration that shown reward may influence hit rate and decision-making strategies. In the congruent group we hypothesized to replicate the effect of reward on hit rate from the previous experiment to ensure that the presentation of reward information during the test phase per se does not influence performance in the Motivated Learning Task. Reward levels were 50, 750, 1450 and 2150 gems, and exposure durations were 1500, 1833, 2167 and 2500 ms.

Experiment 3 ($N = 187$; we aimed to recruit 200 participants, however, a coding error led to additional exclusions after finalizing data collection) mainly corresponded to the congruent group of experiment 2 (i.e., corresponding rewards were presented during learning and test, and confidence ratings in the old/new decision were collected), but was changed in key features. First, we changed the stimulus material. In an effort to make the task easier for participants and increase overall memory for the task contents, we used more distinct pictures of interior scenes, exterior scenes and buildings (which were more similar to Adcock et al.[2] and previously used in Feld et al.[8] and Alizadeh Asfestani et al.[25]). Furthermore, we used two instead of four exposure levels (1500 and 2000 ms), and two instead of four reward levels (also more similar to previous research). Most importantly, reward was changed to monetary values instead of gems/points (£ 0.20 or £ 10). Since images of money should act as a secondary reinforcer that has already been strongly learned by participants, we assumed that this would maximize any cue related dopamine reaction[27]. The entire study was still embedded into the pirate cover story, but instead of gem-filled treasure chests, participants saw pictures of the monetary reward before each stimulus during the learning and test phase (see Fig. 1). These changes were implemented to increase the effect of rewards on memory strength found in experiment 2. At the same time, experiment 3 served as a replication.

## Stanford Sleepiness Scale (SSS)

The SSS is a subjective measure of an individual's current level of sleepiness[22]. Participants rate their sleepiness on a 7-point scale ranging from "1 - Feeling active and vital; alert; wide awake" to "7 - Almost in reverie; sleep onset soon; lost struggle to remain awake". Low scores indicate a low level of current sleepiness, whereas high scores indicate a greater level of current sleepiness.

## Psychomotor Vigilance Task (PVT)

The PVT is a sustained attention task, which measures an individual's objective vigilance. We used a 3 min version of the PVT[23]. In this reaction time task, participants pressed the space bar as soon as a millisecond clock appeared on the screen. The following measures were analyzed: median reaction speed (1/reaction time in ms) and percentage of lapses (number of lapses (reaction time ≥ 500 ms) divided by the number of valid stimuli, excluding premature responses (reaction time ≥ 100 ms)). Reaction times <100 ms were regarded as premature responses and treated as errors of commission. These analysis thresholds are based on Basner & Dinges[28]; note that exclusion criteria for participants were more lenient (see Supplementary Table 2).

## Statistics and reproducibility

All *t*-tests are reported as Welch's *t*-test, which does not assume equal variance. Effect sizes are reported as Cohen's *d*. (Generalized) linear mixed models were calculated using the R package lme4 (version 1.1.34[29]). Unless reported otherwise, two-sided tests are reported at $\alpha = .05$. We report *p*-values for these models using Satterthwaite's degrees of freedom method. Unless reported otherwise, the linear mixed models were run on the trial-wise (i.e., unaggregated) data. For linear mixed models, *t*-values and degrees of freedom are reported. For generalized linear mixed models, *z*-values and the number of observations are reported, and $\beta$-weights are reported in log-odds. In experiment 1 and 2, the predictor reward was transformed as reward / 1000, expressing any effects of reward in units of 1000 gems. Likewise, the predictor duration was transformed as duration / 1000, expressing any effects of duration in units of 1 s. Confidence was coded as 0 = "guess", 1 = "sure", 2 = "very sure". Image type was coded as lure = -0.5 and target = 0.5. In experiment 2, group was coded as incongruent = -0.5 and congruent = 0.5. In our model equations, (…|participant) and (…|image) denote random effects by participant and individual image, respectively (for details on model notation, see Singman & Kellen[30]). For each analysis, we first tried to fit a maximal model, as recommended by Barr et al.[31]. When a model did not converge or resulted in a singular fit, we reduced it by first removing correlations between random slopes and intercepts. If convergence was still not achieved, we next removed random slopes. The full outputs for each model are reported in the Supplementary Note 7. Sensitivity ($d'$) was calculated as $z$(hit rate)-$z$(false alarm rate) and criterion $C$ was calculated as $-.5*(z$(hit rate)$+z$(false alarm rate)) according to Macmillan & Creelman[19]. A reviewer suggested drift diffusion models as an alternative analysis strategy, because drift diffusion models take into account reaction times. This allows for a quantification of "cautiousness" (in form of the parameter "boundary separation"), which could e.g. be expected to increase with reward. Drift diffusion models (DDM) were fit separately for each participant. Given that maximum-likelihood methods for estimating DDM parameters can be particularly sensitive to outliers, we opted to exclude trials outside the 150 ms – 5000 ms range. Furthermore, only participants with a minimum of 10 trials per condition were considered[32]. For each model, we compared a complex model to a simple model. For the simple model, $\alpha$ (boundary separation), $\tau$ (non-decision time), $\beta$ (starting point or bias) and $\delta$ (drift rate) were estimated based on the entire data range, regardless of any variables of interest. In the complex model, we let the parameters $\alpha$, $\beta$, $\delta$, and $\tau$ differ by a variable of interest (e.g., reward; see respective model descriptions), resulting in an $\alpha$, $\beta$, $\delta$, and $\tau$ estimate for every level of the variable of interest per participant. Model comparisons were based in the BIC (Bayesian Information Criterion) and AIC (Akaike Information Criterion), where we regarded the complex model to be preferable when the difference in information criteria (simple model - complex model) was >0. Additionally, we compared the two models in a likelihood ratio test. This likelihood ratio test is based on a $\chi^2$ distribution, where the degrees of freedom are determined by the difference in the number of parameters between the two models. A statistically significant likelihood ratio test indicates that the fit of the complex model is better. Comparisons between the two models were conducted on the participant level, and we report the percentage of cases where the complex model was preferred over the simple model. The full outputs of the drift diffusion analyses and plots of the results are reported in the Supplementary Note 9.

**Fig. 2 | Hits and false alarms in experiment 1. a** Hit rate for each reward level in experiment 1, based on $N = 200$ participants. The hit rate significantly increased as the reward increased. In the beeswarm plot, orange dots represent the hit rate per reward level for each participant. Light grey lines connect the dots belonging to the same participant. The line plot shows the mean hit rate for each reward level. Black error bars show the within-subject standard error, as implemented in the R package Rmisc[50,51]. The *p*-value for the reward effect in the mixed model from the main text is reported. Asterisks represent significance at $\alpha = .05$. **b** Number of absolute hits and false alarms per confidence level and reward level for trials where participants said a picture was old. The number of trials on the y-axis represents the sum across all $N = 200$ participants (bars are stacked). **c** The absolute difference in the number of hits and false alarms (summed across all $N = 200$ participants) for each expected reward level and confidence level. We found that only when confidence is high, false alarms relative to hits were reduced the higher the expected reward was. *p*-values for the effect of expected reward in three generalized linear mixed models (outcome: false alarm (1) or not (0)) per confidence level are reported (see main text). Asterisks represent significance at $\alpha = .05$.

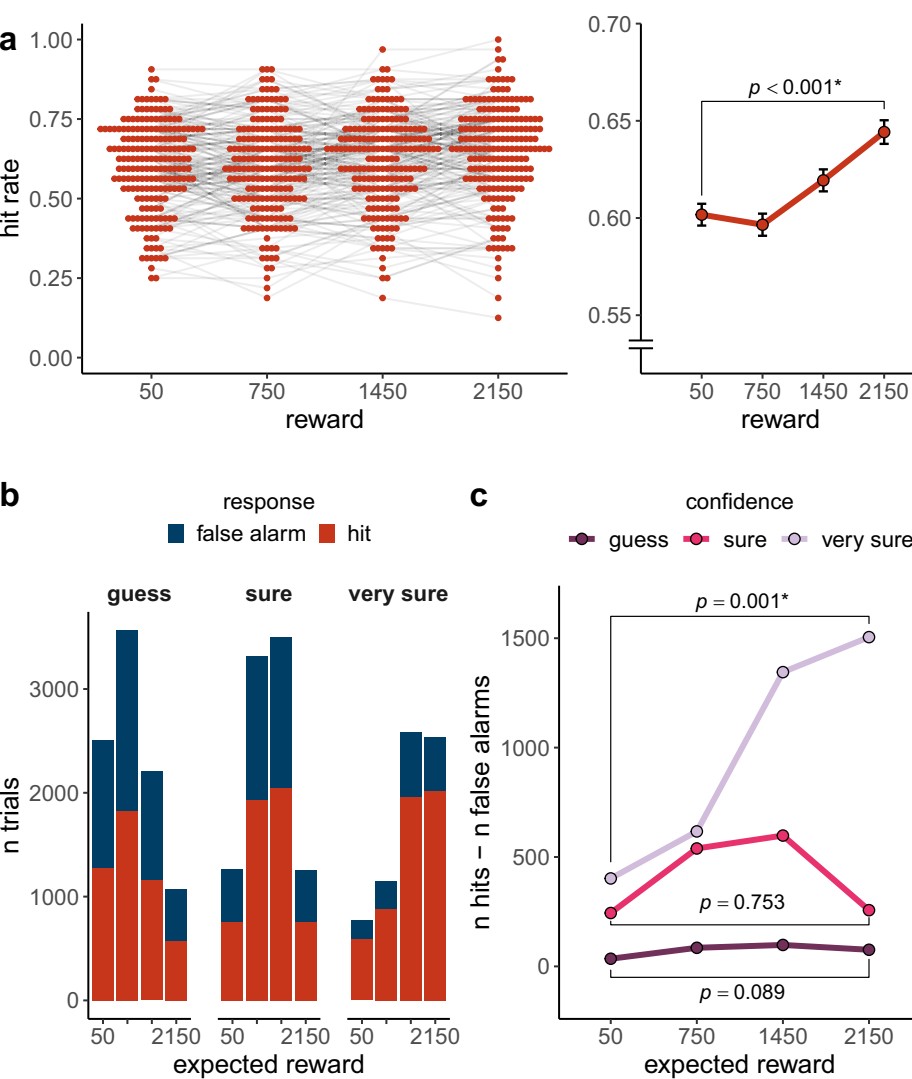

## Reporting summary

Further information on research design is available in the Nature Portfolio Reporting Summary linked to this article.

## Results

### Experiment 1

To evaluate the effect of reward on hits, we ran the following generalized linear mixed model with a logit link function: hit (0 or 1) ~ reward + ( reward | participant) + (1 | image) (see Supplementary Table 11; we report the effect of adding an interaction with exposure duration in Supplementary Table 48). There was a statistically significant increase in hits the higher the reward was, $\beta = 0.11$, $SE = 0.02$, $z(25600) = 5.59$, $p < 0.001$ (see Fig. 2a). Consequently, the hit rate for the highest reward level ($M = 0.64$, $SD = 0.15$) was significantly higher than for the lowest reward level ($M = 0.60$, $SD = 0.14$), $t(199) = 5.00$, $p < 0.001$, $d_z = 0.35$, 95% CI [0.21, 0.50]. We explored the effect of reward, confidence and their potential interaction in the following generalized linear mixed model with a logit link function: hit (0 or 1) ~ reward * confidence + (reward + confidence || participant) + (confidence |image) (see Supplementary Table 12). There was no statistically significant effect of reward on hits, $\beta = 0.05$, $SE = 0.03$, $z(25600) = 1.67$, $p = 0.096$, but a statistically significant increase in hits as confidence increased $\beta = 0.45$, $SE = 0.07$, $z(25600) = 6.36$, $p < 0.001$. Additionally, there was a statistically significant interaction between reward and confidence, $\beta = 0.06$, $SE = 0.02$, $z(25600) = 2.70$, $p = 0.007$. To follow up on the

interaction, we ran the following generalized linear mixed model with a logit link function: hit (0 or 1) ~ reward + (1 | participant) + (1 | image), separately for each confidence level. Reward had no statistically significant effect on hits for the lowest confidence level ("guess"), $\beta = 0.05$, $SE = 0.03$, $z(8827) = 1.77$, $p = 0.076$ (see Supplementary Table 13). However, when participants were "sure", there was a statistically significant increase in hits the higher the reward, $\beta = 0.09$, $SE = 0.03$, $z(8657) = 2.85$, $p = 0.004$ (see Supplementary Table 14), or "very sure", $\beta = 0.19$, $SE = 0.04$, $z(8116) = 4.86$, $p < 0.001$ (see Supplementary Table 15).

Since lures do not have a true reward, we evaluated the influence of the expected reward on false alarms. That is, for trials where participants said that an image was "old", we ran the following generalized linear mixed model with a logit link function: false alarm (0 or 1) ~ expected reward * confidence + (expected reward || participant) + (1 | image) (see Supplementary Table 16). There was a statistically significant decrease in false alarms relative to hits as confidence increased, $\beta = -0.49$, $SE = 0.04$, $z(25711) = -13.81$, $p < 0.001$. There was no statistically significant main effect of expected reward on false alarms, $\beta = 0.01$, $SE = 0.03$, $z(25711) = 0.31$, $p = 0.755$, but a statistically significant interaction between expected reward and confidence, $\beta = -0.10$, $SE = 0.03$, $z(25711) = -3.73$, $p < 0.001$ (see Fig. 2b and c). We further investigated this interaction by fitting the following generalized linear mixed model with a logit link function for each confidence level separately: false alarm (0 or 1) ~ expected reward + (1 | participant). For the lowest confidence level ("guess"), there

**Fig. 3 | Reward expectations in experiment 1.**
**a** Expected reward per level of true reward and confidence for hits, based on N = 200 participants. We found no statistically significant main effect of true reward on reward expectations, but the higher participants' confidence, the higher their reward expectations (main effect of confidence: *p*<.001). The relationship between true reward and expected reward varied by confidence (reward x confidence interaction: *p*<.001): When participants were confident, the true reward associated with a target predicted reward expectations. In the beeswarm plot, coloured dots (jittered randomly along the *x*-axis) represent the mean expected reward per true reward category for each individual participant. The line plot shows the group means. Black error bars show the within-subject standard error, as implemented in the R package Rmisc[50,51]. *P*-values for the reward effect in three mixed models per confidence level are reported (see main text). Asterisks represent significance at α = .05. **b** Expected reward per level of confidence for false alarms, based on N = 200 participants. The higher participants' confidence, the higher the reward they expect for false alarms. Coloured dots show the expected reward for each confidence level. **c** The correlation between the reward effect on the hit rate (in the model hit ~ reward + (reward | participant) + (reward | image)) and the percentage of correct reward expectations. The more correct reward expectations are, the stronger the effect of true reward on hit rate. **d** The correlation between the reward effect on the hit rate (in the model hit ~ reward + (reward | participant) + (reward | image)) and the reward effect on expected reward (in the model expected reward ~ (reward | participant)). The more accurate the relationship between true reward and expected reward, the stronger the relationship between reward and hit rate. Pearson correlations each based on N = 200 participants are reported. Asterisks represent significance at α = 0.05.

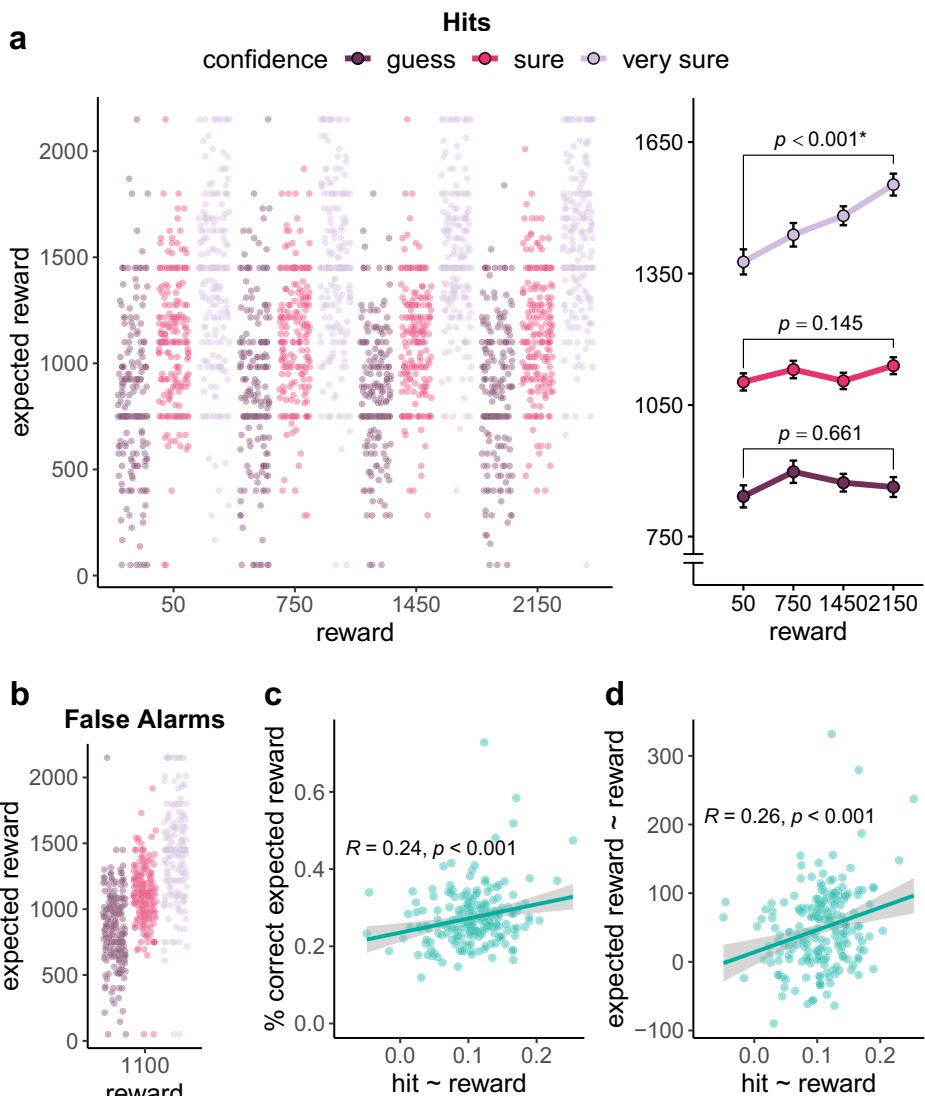

was no statistically significant effect of expected reward on false alarms, $\beta = -0.05$, $SE = 0.03$, $z(9356) = -1.70$, $p = .089$ (see Supplementary Table 17). Likewise, there was no statistically significant effect of expected reward on false alarms for the medium confidence level ("sure"), $\beta = -0.01$, $SE = 0.03$, $z(9328) = -0.31$, $p = .753$ (see Supplementary Table 18). However, for the highest confidence level ("very sure"), there was a statistically significant reduction in false alarms relative to hits as reward expectations increased, $\beta = -0.15$, $SE = 0.04$, $t(7027) = -3.27$, $p = .001$ (see Supplementary Table 19). We furthermore investigated the distribution of the number of trials per expected reward level. For false alarms, participants did not expect each reward level equally often, $\chi^2(3) = 1002.01$, $p$<.001 (see Supplementary Fig. 1).

We evaluated the effect of reward, confidence and their interaction on the reward participants expected to receive in the following linear mixed model run on hit trials only: expected reward ~ reward * confidence + ( confidence * reward || participant) (see Supplementary Table 20). We found no statistically significant association between the amount of reward associated with the target and how much reward participants expected to receive when they identified the target as old, $\beta = -8.01$, $SE = 11.36$, $t(1108.19) = -0.71$, $p = .480$, but there was a statistically significant effect of confidence: Participants expected higher rewards the higher their confidence was, $\beta = 241.64$, $SE = 13.60$, $t(703.98) = 17.70$, $p$<.001. Crucially, the higher participants' confidence, the higher the relationship

between the true reward and the reward participants expected to receive, $\beta = 45.37$, $SE = 8.64$, $t(1249.44) = 5.25$, $p$<.001 (see Fig. 3a). Following up on the interaction, we evaluated the effect of true rewards on expected rewards individually for the different confidence levels with the following linear mixed model run on hit trials only: expected reward ~ reward + ( reward || participant). For high-confidence hits we found a statistically significant association between true reward and expected reward $\beta = 90.97$, $SE = 12.32$, $t(340.10) = 7.39$, $p$<.001 (see Supplementary Table 23). This was not the case for any of the other confidence levels (all $p > 0.661$; see Supplementary Tables 21 and 22). We investigated the effect of confidence on reward expectations for false alarm trials with the following linear mixed model: expected reward ~ confidence + (confidence || participant) (see Supplementary Table 24). Participants likewise expected to receive a higher reward with increasing confidence $\beta = 272.60$, $SE = 12.70$, $t(207.23) = 21.46$, $p$<.001 (Fig. 3b).

We next explored whether more accurate reward expectations were related to a larger effect of true reward on the hit rate. To this end, we correlated the per-participant slopes for the reward effect from the hit ~ reward model with the % of correct reward expectations for hits per participant. This quantifies the relationship between the proportion of accurate reward expectations (binary classification) and how strongly the hit rate is influenced by the true reward. For participants whose reward expectations were more accurate, the hit rate was more strongly influenced by the true

https://doi.org/10.1038/s44271-024-00074-9 **Article**

reward, $r(198) = .24$, $p<.001$ (Fig. 3c). Additionally, we calculated the correlation with the per-participant slopes for the reward effect from the hit ~ reward model with the per-participant slopes for the reward effect from an expected reward ~ reward model (expected reward ~ reward + (reward| participant) for hits. Other than the previous analysis, this analysis does not only focus on correct reward expectations, but acknowledges that a participant expecting 1450 when the true reward is 2150 is closer to the truth than a participant expecting 50 gems. Participants whose reward expectations were more strongly influenced by the true reward showed a larger influence of reward on the hit rate, $r(198) = .26$, $p<.001$ (see Fig. 3d).

To investigate whether exposure duration influences hit rate, we ran the following generalized linear mixed model with a logit link function: hit (0~ or 1) ~ duration + (duration || participant) + (1 | image) (see Supplementary Table 25). The longer an image was presented during the learning phase, the more likely it was that participants made a hit in a given target trial, $\beta = 0.18$, $SE = 0.04$, $z(25600) = 4.72$, $p<.001$. Consequently, the hit rate for the longest exposure duration of 2500 ms ($M = 0.63$, $SD = 0.15$) was significantly higher than for the shortest exposure duration of 1500 ms ($M = 0.59$, $SD = 0.14$), $t(199) = 4.21$, $p<.001$, $d_z = 0.30$, 95% CI [0.16, 0.44].

In an exploratory alternative analysis, we investigated whether the expected reward had an effect on whether participants made a hit or a false alarm. That is, we ran a drift diffusion model on all trials where participants responded that a picture was "old" (i.e., where data about reward expectations was available), with the possible outcomes "hit" (upper boundary) or "false alarm" (lower boundary). For the complex model, we let boundary separation, starting point, drift rate, and non-decision time vary by expected reward level (50, 750, 1450 or 2150 gems). After excluding participants with <10 trials per condition, the model was run on $n = 161$ participants. According to the BIC, the complex model was preferred in 100.00% of cases. According to the AIC, the complex model was preferred in 100.00% of cases. According to the number of statistically significant likelihood ratio tests, the complex model provided a better fit in 49.69% of cases.

To estimate the effect of reward on boundary separation $\alpha$, we ran a linear mixed model of the form alpha ~ expected reward + (expected reward | participant) (see Supplementary Table 51). There was no statistically significant effect of reward on boundary separation, $\beta = -0.02$, $SE = 0.03$, $t(160.00) = -0.72$, $p = .472$ (see Supplementary Fig. 5a). That is, there was no statistically significant difference in boundary separation between the highest expected reward level ($M = 2.04$, $SD = 0.89$) and the lowest expected reward level ($M = 2.06$, $SD = 0.39$), $t(160) = -0.28$, $p = .777$, $d_z = -0.02$, 95% CI [−0.18, 0.13]. To estimate the effect of reward on the starting point $\beta$, we ran a linear mixed model of the form beta ~ expected reward + (expected reward | participant) (see Supplementary Table 52). The association between the expected reward and the starting point was not statistically significant, $\beta = 0.00$, $SE = 0.01$, $t(160.00) = -0.65$, $p = .513$ (see Supplementary Fig. 5b). That is, there was no statistically significant difference between the starting point for the highest expected reward level ($M = 0.52$, $SD = 0.12$) and the starting point for the lowest expected reward level ($M = 0.53$, $SD = 0.11$), $t(160) = -0.75$, $p = .455$, $d_z = -0.06$, 95% CI [−0.21, 0.10]. However, a linear mixed model of the form delta ~ expected reward + (expected reward | participant) (see Supplementary Table 53) revealed that the drift rate increased as the expected reward increased, $\beta = 0.17$, $SE = 0.02$, $t(160.00) = 8.27$, $p<.001$ (see Supplementary Fig. 5c). That is, the drift rate for the highest expected reward level ($M = 0.45$, $SD = 0.48$) was significantly higher than the drift rate for the lowest expected reward level ($M = 0.11$, $SD = 0.35$), $t(160) = 7.52$, $p<.001$, $d_z = 0.59$, 95% CI [0.42, 0.76]. A linear mixed model of the form tau ~ expected reward + (expected reward | participant) (see Supplementary Table 54) showed a trend towards a decreasing non-decision time as the reward increased, $\beta = -0.02$, $SE = 0.01$, $t(160.00) = -1.88$, $p = .062$ (see Supplementary Fig. 5d). That is, the non-decision time for the highest expected reward level ($M = 1.04$, $SD = 0.24$) was significantly lower than for the lowest expected reward level ($M = 1.08$, $SD = 0.27$), $t(160) = -2.00$, $p = .048$, $d_z = -0.16$, 95% CI

[−0.31, 0.00]. In general, participants median reaction time (in seconds) was shorter when they expected a high ($M = 1.76$, $SD = 0.39$) versus low reward ($M = 1.94$, $SD = 0.52$), $t(160) = -7.50$, $p<.001$, $d_z = -0.59$, 95% CI [−0.76, −0.42].

## Experiment 2

To evaluate the effect of reward on hits in the congruent group, we ran the following generalized linear mixed model with a logit link function: hit (0 or 1) ~ reward + (reward | participant) + (reward | image) (see Supplementary Table 26; we report the effect of adding an interaction with exposure duration in Supplementary Table 49). There was a statistically significant increase in hits the higher the reward was, $\beta = 0.13$, $SE = 0.03$, $z(13184) = 4.94$, $p<.001$ (see Fig. 4a). Consequently, the hit rate for the highest reward level ($M = 0.63$, $SD = 0.12$) was significantly higher than for the lowest reward level ($M = 0.56$, $SD = 0.13$), $t(102) = 5.25$, $p<.001$, $d_z = 0.52$, 95% CI [0.31, 0.72]. Even though this effect of reward on hits is nominally larger than in experiment 1, a comparison between the two experiments (hit (0 or 1) ~ reward * experiment + (reward + experiment || participant) + (1 | image); see Supplementary Table 27), did not show a statistically significant reward $x$ experiment interaction, $\beta = 0.04$, $SE = 0.03$, $z(38912) = 1.33$, $p = .182$.

We analyzed the effect of true and shown reward on hits in the incongruent group, with the following generalized linear mixed model with a logit link function: hit (0 or 1) ~ true reward + shown reward + (shown reward | participant) + (1 | image) (see Supplementary Table 28). The higher the shown reward was the more hits participants achieved, $\beta = 0.15$, $SE = 0.04$, $z(13056) = 4.05$, $p<.001$ (see Fig. 4c), but the true reward had no statistically significant effect, $\beta = 0.02$, $SE = 0.02$, $z(13056) = 0.96$, $p = .337$ (see Fig. 4b). That is, the hit rate for the highest shown reward level (2150 gems) ($M = 0.62$, $SD = 0.17$) was higher than for the lowest shown reward level (50 gems) ($M = 0.56$, $SD = 0.16$), $t(101) = 3.80$, $p<.001$, $d_z = 0.38$, 95% CI [0.17, 0.58]. However, there was no statistically significant difference between the hit rate for the highest true reward level ($M = 0.60$, $SD = 0.13$) and the lowest true reward level ($M = 0.60$), $SD = 0.15$, $t(101) = 0.08$, $p = .936$, $d_z = 0.01$, 95% CI [−0.19, 0.20]. Even though the effect of reward on hits was nominally larger in the congruent group than the effect of shown reward on hits in the incongruent group, a comparison between the two groups (hit (0 or 1) ~ shown reward * group + (shown reward + group | participant) + (1 | image); see Supplementary Table 29) did not show a statistically significant reward $x$ experiment interaction, $\beta = -0.01$, $SE = 0.04$, $z(26240) = -0.13$, $p = .895$.

In an exploratory analysis, we investigated whether the influence of the shown reward on participants' decisions was associated with their overall memory performance. First, we extracted the by-participant slope for the shown reward effect from a mixed model of the form response (old or new) ~ shown reward + (shown reward | participant) + (shown reward | image). We then correlated the (absolute) slope with participants' overall memory strength in the task ($d'$). Participants whose old/new decisions were more strongly influenced by the shown reward (i.e., the more the slope differed from 0) performed worse in the Motivated Learning Task, $r(100) = -.31$, $p = .002$ (see Fig. 4d). Upon visual inspection, it seemed like only a few data points belonging to participants with steeper slopes were driving the correlation. However, when removing those outliers (all participants with a slope $\geq .3$), the correlation remained statistically significant, $r(89) = -.23$, $p = .030$. The shown reward might influence target trials differently, because for targets, shown reward information can potentially be in conflict with the true reward information. This is why we repeated the previous exploratory analysis for target trials. We extracted the by-participant slope for the shown reward effect from a mixed model of the form hit (0 or 1) ~ shown reward + (shown reward | participant) + (1 | image). We then correlated the (absolute) slope with participants' overall memory strength in the task ($d'$). Participants whose hits were more strongly influenced by the shown reward (i.e., the more the slope differed from 0) performed worse in the Motivated Learning Task, $r(100) = -.29$, $p = .003$ (see Fig. 4e). Upon visual inspection, it seemed like only a few data points belonging to

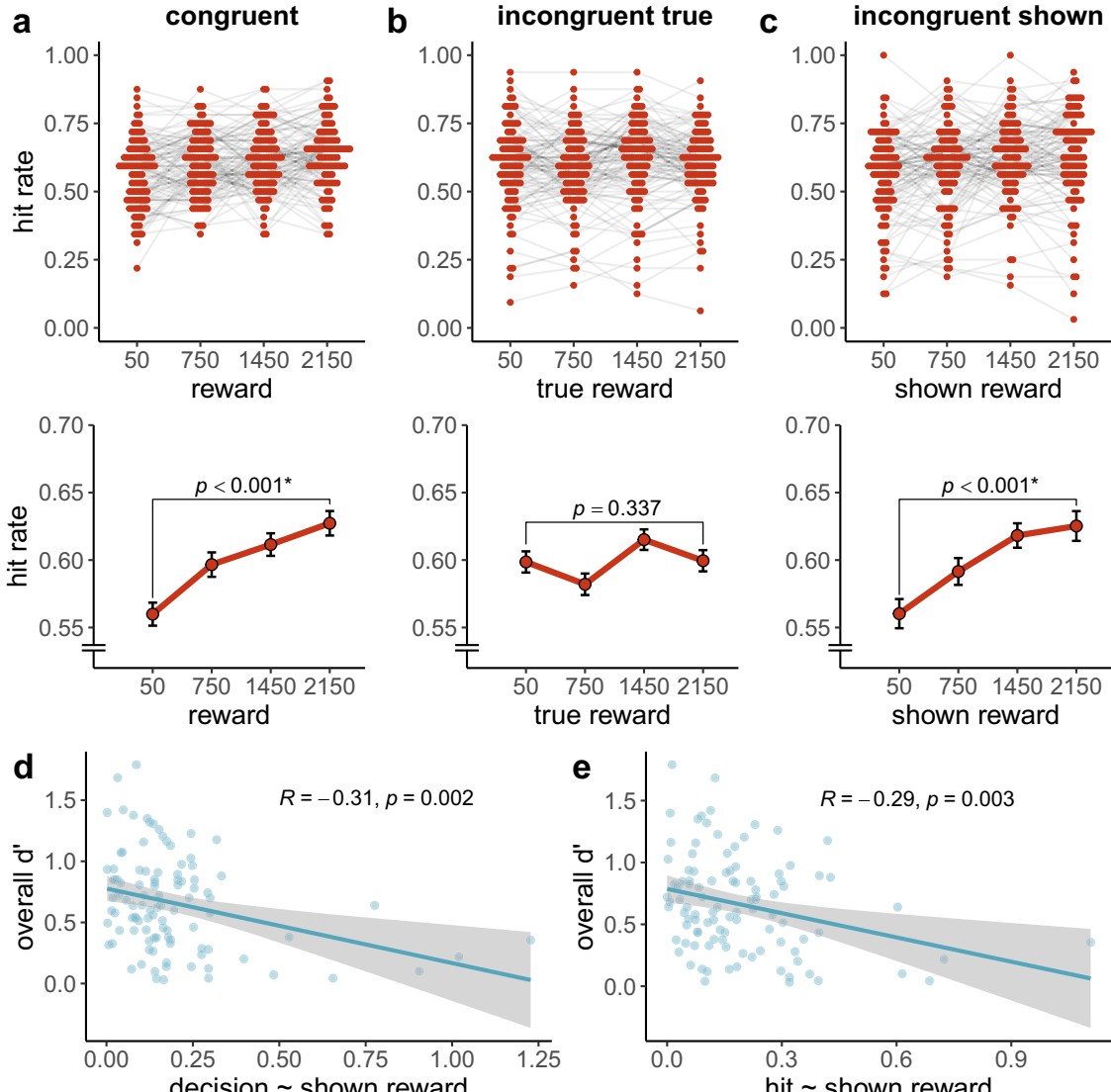

**Fig. 4 | Hit rate in experiment 2. a** Hit rate per reward level for the congruent group (where true and shown reward were identical) based on $n = 103$ participants. Participants' hit rate was higher the higher the reward was. **b** Hit rate per level of true reward for the incongruent group based on $n = 102$ participants. There was no statistically significant association between true reward and hit rate. **c** Hit rate per level of shown reward for the incongruent group based on $n = 102$ participants. Participants' hit rate was higher the higher the shown reward was. For each panel the beeswarm plot shows orange dots for each individual participant. Light grey lines connect the dependent data points. The lineplot plot of each panel shows the mean hit rate per reward level. Black error bars show the within-subject standard error, as implemented in the R package Rmisc[50,51]. *P*-values are reported for reward effect in the mixed models reported in text, with asterisks representing significance at $\alpha = 0.05$. **d** The correlation between the shown reward effect on the old/new decision (in

the model decision ~ shown reward + (shown reward | participant) + (shown reward | image)) and overall memory strength in the Motivated Learning Task. The analysis is based on the incongruent group (based on $n = 102$ participants). The stronger the influence of shown reward on participants' decisions, the worse their memory strength. The correlation remained statistically significant when removing participants with steep slopes (slope ≥ .3). **e** The correlation between the shown reward effect on hits (in the model hit ~ reward + (reward | participant) + (1 | image)) and overall memory strength in the Motivated Learning Task. The analysis is based on the incongruent group (based on $n = 102$ participants). The stronger the influence of shown reward on whether participants made a hit or not, the worse their memory strength. The correlation remained statistically significant when removing participants with steep slopes (slope ≥ .5). Pearson correlations are reported. Asterisks represent significance at $\alpha = .05$.

participants with steeper slopes were driving the correlation. However, when removing those outliers (all participants with a slope ≥ .5), the correlation remained statistically significant, $r(95) = -.20$, $p = .047$.

We analyzed the false alarms as a function of shown reward and group in the following generalized linear mixed model with a logit link function: false alarm (0 or 1) ~ reward * group + (reward | participant) + (reward || image), run on lure trials only (see Supplementary Table 30). The higher the shown reward was, the more false alarms participants made, $\beta = 0.13$, $SE = 0.03$, $z(26240) = 4.92$, $p<.001$. There was no statistically significant difference in the number of false alarms between the two groups, $\beta = 0.14$, $SE = 0.10$, $z(26240) = 1.43$, $p = .153$, but a statistically significant

interaction between shown reward and group, $\beta = -0.14$, $SE = 0.05$, $z(26240) = -2.76$, $p = .006$ (see Fig. 5). We followed up on the interaction with separate models per group (false alarm (0 or 1) ~ reward + (reward | participant) + (1 | image)). In both the congruent group, $\beta = 0.07$, $SE = 0.03$, $z(13184) = 2.15$, $p = .031$ (see Supplementary Table 31), and the incongruent group, $\beta = 0.20$, $SE = 0.04$, $z(13056) = 4.61$, $p<.001$ (see Supplementary Table 32), the higher the reward for lures, the more false alarms participants made. However, this relationship was stronger for the incongruent group. There was no statistically significant difference in the false alarm rate for the lowest reward level (50 gems) between the incongruent group ($M = 0.31$, $SD = 0.13$), and the congruent group,

**Fig. 5 | False alarm rate in experiment 2.** False alarm rate is shown per level of shown reward for each group (congruent: $n = 103$ participants; incongruent: $n = 102$ participants). In both groups, the false alarm rate increased as the shown reward increased (main effect: $p<.001$). However, this relationship was stronger in the incongruent group (interaction effect: $p = .006$). In the beeswarm plots, blue dots show the false alarm rate per participant, with light grey lines connecting the dependent data points. The line plots panels show the mean false alarm rate per shown reward level. Black error bars show the within-subject standard error, as implemented in the R package Rmisc[50,51]. *P*-values for the effect of reward in a mixed model per group each are reported (see main text). Asterisks represent significance at $\alpha = .05$.

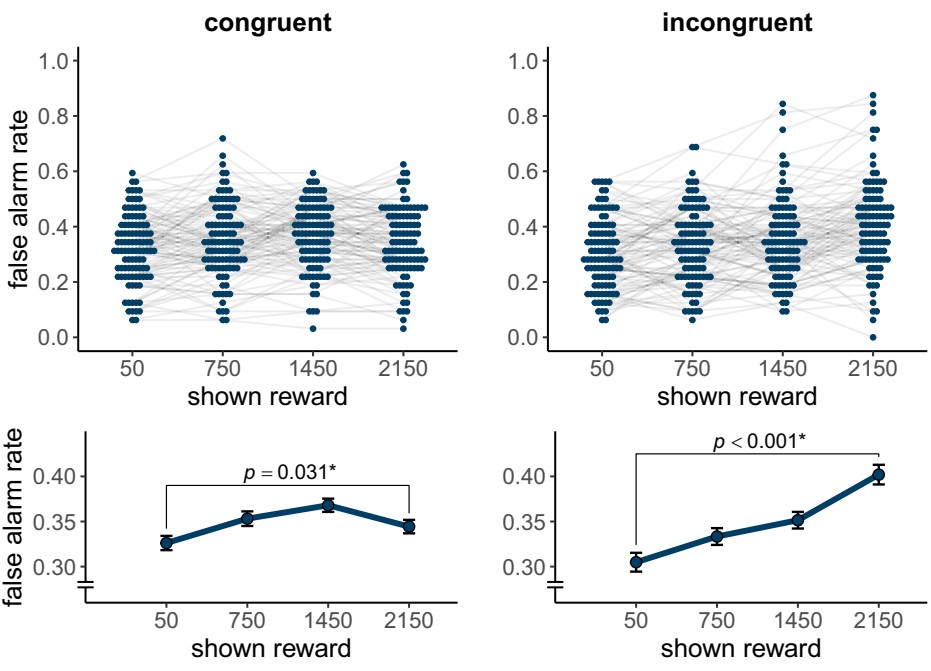

$(M = 0.32, SD = 0.12)$, $t(202.39) = -1.07$, $p = .288$, $d = -0.15$, $95\%$ CI $[-0.42, 0.13]$. But for the highest reward level (2150 gems), the false alarm rate was significantly higher for the incongruent group $(M = 0.40, SD = 0.16)$, than for the congruent group, $(M = 0.34, SD = 0.12)$, $t(189.55) = 2.97$, $p = .003$, $d = 0.42$, $95\%$ CI $[0.14, 0.69]$.

Descriptively, the false alarm rate drops between the 3rd (1450 gems; $M = 0.37, SD = 0.12$) and highest (2150 gems; $M = 0.34, SD = 0.12$) reward level, but exploring this difference, we found that it was not statistically significant, $t(203.45) = -1.40$, $p = .163$, $d = -0.19$, $95\%$ CI $[-0.47, 0.08]$.

For the congruent group, we preregistered an effect of reward on memory strength, measured as d'. We tested this hypothesis using the following linear mixed model: d' ~ reward + (reward || participant). We found that there was a statistically significant increase in d' the higher the reward was, $\beta = 0.06$, $SE = 0.02$, $t(140.67) = 2.54$, $p = .012$ (see Fig. 6a and Supplementary Table 33). Accordingly, d' for the highest reward level (2150 gems) $(M = 0.79, SD = 0.51)$ was significantly higher than for the lowest reward level (50 gems) $(M = 0.66, SD = 0.43)$, $t(102) = 2.85$, $p = .005$, $d_z = 0.28$, $95\%$ CI $[0.08, 0.48]$. ROC analyses corroborated these findings (see Supplementary Fig. 2). Upon visual inspection (see Fig. 6a), it appeared that the effect of reward on d' seemed to be mainly driven by the highest reward level, while there seemed to be no difference between the lower levels. We followed this up with an additional exploratory linear mixed model of the form d' ~ reward + (1 | participant), where reward was not treated as a linear, but a categorical predictor (because more parameters must be estimated for a categorical predictor, the model structure had to be simplified in order to reach convergence). The lowest reward level was used as reference category. Indeed, only for the highest reward level there was a statistically significant difference from the first level $\beta = 0.13$, $SE = 0.05$, $t(306.00) = 2.90$, $p = .004$, but not for the two medium reward categories (all $p> .669$; see Supplementary Table 34).

For the incongruent group, a meaningful d' could only be calculated for the shown reward. We preregistered that there would be no statistically significant effect of shown reward on d'. In line with this, a linear mixed model (d' ~ shown reward + (shown reward | participant); see Supplementary Table 35) did not show a statistically significant effect of shown reward on d', $\beta = -0.04$, $SE = 0.02$, $t(101.00) = -1.61$, $p = .110$ (Fig. 6b). Accordingly, there was no statistically significant difference in d' between the highest shown reward level (2150 gems) $(M = 0.61$,

$SD = 0.54)$ and the lowest shown reward level (50 gems) $(M = 0.71, SD = 0.50)$, $t(101) = -1.72$, $p = .088$, $d_z = -0.17$, $95\%$ CI $[-0.37, 0.03]$.

To investigate the effect of reward on decision-making (criterion $C$), we ran the following linear mixed model: criterion ~ reward + (reward | participant), in the congruent group (see Supplementary Table 36). The higher the reward was, the more lenient criterion was (i.e., participants were more likely to categorize a picture as old for higher rewards), $\beta = -0.06$, $SE = 0.01$, $t(101.99) = -4.84$, $p<.001$ (see Fig. 6c). Criterion for the lowest reward level $(M = 0.16, SD = 0.29)$ was more strict than for the highest reward level $(M = 0.04, SD = 0.26, t(102) = -4.56, p<.001, d_z = -0.45, 95\%$ CI $[-0.65, -0.25]$. For the incongruent group, a meaningful criterion could only be calculated for the shown reward. We preregistered that there might be an effect of shown reward on criterion. In a linear mixed model (criterion ~ shown reward + (shown reward | participant); see Supplementary Table 37), we found that the criterion became more lenient the higher the shown reward was $\beta = -0.11$, $SE = 0.02$, $t(101.00) = -5.19$, $p<.001$ (see Fig. 6d). Criterion for the lowest reward level $(M = 0.20, SD = 0.35)$ was more strict than for the highest reward level $(M = -0.03, SD = 0.42)$, $t(101) = -5.24$, $p<.001$, $d_z = -0.52$, $95\%$ CI $[-0.72, -0.31]$.

We explored whether confidence affected hits, potentially interacting with reward, in the following generalized linear mixed model with a logit link function for the congruent group: hit (0 or 1) ~ reward * confidence + ( reward + confidence | participant) + (confidence | image) (see Supplementary Table 38). The higher their confidence, the more hits participants made, $\beta = 0.55$, $SE = 0.10$, $z(13184) = 5.66$, $p<.001$. However, there was no statistically significant effect of reward on hits, $\beta = 0.08$, $SE = 0.04$, $z(13184) = 1.69$, $p = .090$ (although the *p*-value suggests a trend in the same direction as without adding confidence), and no statistically significant interaction between reward and confidence, $\beta = 0.05$, $SE = 0.04$, $z(13184) = 1.32$, $p = .187$. For the incongruent group, we explored the influence of true reward and confidence on hits in the following generalized linear mixed model with a logit link function: hit (0 or 1) ~ true reward * confidence + (confidence || participant) + (1 | image) (see Supplementary Table 39). With increasing confidence, hits increased, $\beta = 0.58$, $SE = 0.09$, $z(13056) = 6.53$, $p<.001$, but there was no statistically significant effect of true reward on hits, $\beta = 0.05$, $SE = 0.04$, $z(13056) = 1.22$, $p = .221$, nor a statistically significant interaction between true reward and confidence,

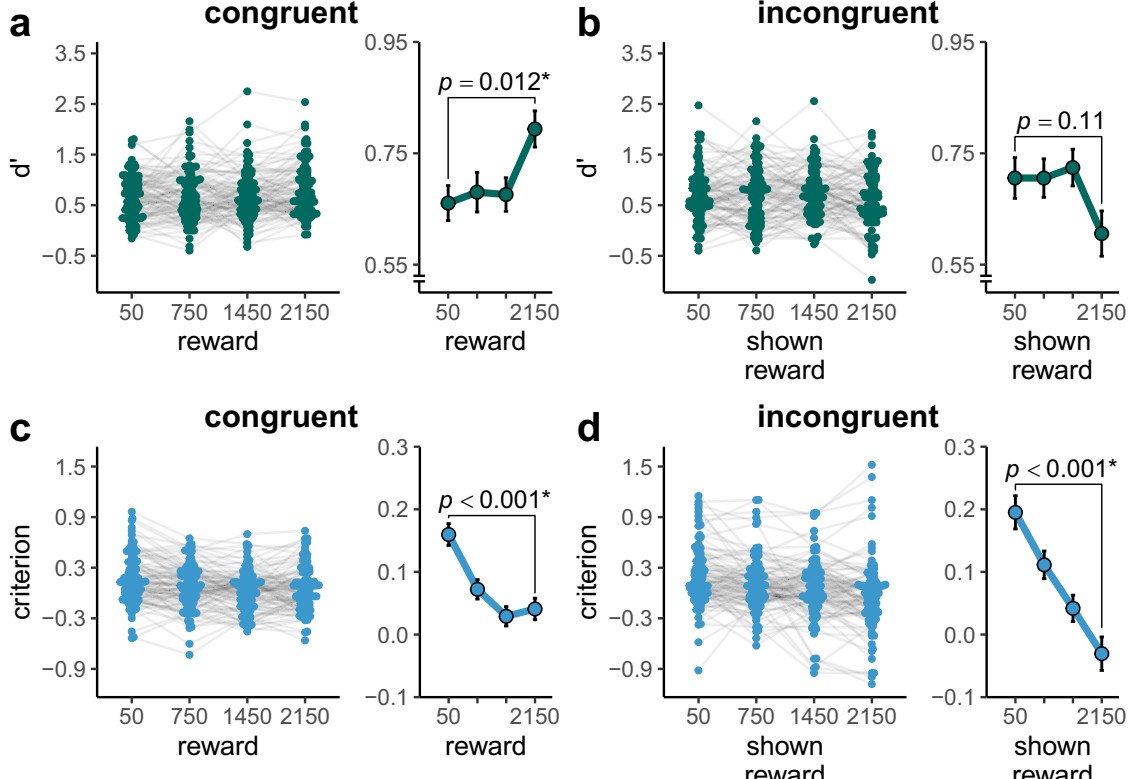

**Fig. 6 | Memory strength and decision criterion in experiment 2. a** Memory strength d' per level of reward for the congruent group. Memory strength increased with increasing reward. **b** Memory strength d' per level of shown reward for the incongruent group. The effect of shown reward on memory strength was not statistically significant. Note that participants with an overall memory performance of d' ≤ 0 were excluded from data analysis. However, participants may still show a performance of d' ≤ 0 when d' is calculated for each reward level. **c** Decision criterion C per level of reward for the congruent group. The higher the reward, the more lenient the decision criterion was. **d** Decision criterion C per level of shown reward

for the incongruent group. The higher the shown reward, the more lenient the decision criterion was. In the beeswarm plots of each panel, coloured dots represent data per participant, with light grey lines connecting the dependent data points. In the line plots of each panel, coloured dots show group means. Black error bars show the within-subject standard error, as implemented in the R package Rmisc[50,51]. For the congruent group, n = 103 participants are shown, for the incongruent group, n = 102 participants are shown. *P*-values for the reward effect in separate models for memory strength and decision criterion per group are reported (see main text). Asterisks represent significance at α = .05.

$\beta = -0.02$, $SE = 0.03$, $z(13056) = -0.57$, $p = .572$. We further investigated the influence of shown reward and confidence on hits in the incongruent group with the following generalized linear mixed model with a logit link function: hit (0 or 1) ~ shown reward * confidence + (shown reward + confidence | participant) + (confidence || image) (see Supplementary Table 40). The higher the shown reward, the more hits participants made, $\beta = 0.19$, $SE = 0.05$, $z(13056) = 3.69$, $p<.001$, and the higher their confidence, the more hits participants made, $\beta = 0.61$, $SE = 0.10$, $z(13056) = 5.99$, $p<.001$. However, there was no statistically significant interaction between shown reward and confidence, $\beta = -0.05$, $SE = 0.04$, $z(13056) = -1.37$, $p = .170$.

We explored whether confidence and reward affected false alarms in the congruent group using the following generalized linear mixed model with a logit link function: false alarm (0 or 1) ~ shown reward * confidence + (shown reward | participant) + (1 | image) (see Supplementary Table 41). With increasing confidence, false alarms decreased, $\beta = -0.31$, $SE = 0.05$, $z(13184) = -5.75$, $p<.001$, but there was no statistically significant effect of shown reward on false alarms, $\beta = 0.05$, $SE = 0.05$, $z(13184) = 1.04$, $p = .300$, nor a statistically significant interaction between shown reward and confidence, $\beta = 0.03$, $SE = 0.04$, $z(13184) = 0.71$, $p = .480$. For the incongruent group, we investigated whether confidence and reward affected false alarms using the following generalized linear mixed model with a logit link function: false alarm (0 or 1) ~ shown reward * confidence + (shown reward + confidence | participant) + (1 | image) (see Supplementary Table 42). The higher the shown reward, the more false alarms participants made, $\beta = 0.20$, $SE = 0.06$, $z(13056) = 3.51$, $p<.001$, and the higher their

confidence, the less false alarms participants made, $\beta = -0.53$, $SE = 0.10$, $z(13056) = -5.52$, $p<.001$. However, there was no statistically significant interaction between shown reward and confidence, $\beta = 0.01$, $SE = 0.04$, $z(13056) = 0.26$, $p = .794$.

By introducing rewards that are orthogonal to the true reward, we might have decreased participants' confidence in their memory. By violating their reward expectations, participants might not have relied on their memory anymore, but only on the reward information that was presented. In an exploratory analysis, we compared confidence for targets and lures between the two groups in the following model: confidence ~ group * image type + (image type | participant) (see Supplementary Table 43). We found confidence to be higher for targets than for lures, $\beta = 0.10$, $SE = 0.01$, $t(203.01) = 9.37$, $p<.001$. However, there was no statistically significant difference in confidence between the two groups, $\beta = 0.05$, $SE = 0.05$, $t(203.00) = 0.91$, $p = .365$, and no statistically significant group x image type interaction, $\beta = -0.01$, $SE = 0.02$, $t(203.01) = -0.66$, $p = .510$.

To investigate whether exposure duration influences hit rate, we ran the following generalized linear mixed model with a logit link function for the congruent group: hit (0 or 1) ~ duration + (duration | participant) + (1 || image) (see Supplementary Table 44). The longer an image was presented during the learning phase, the more likely it became that participants in the congruent group made a hit in a given target trial, $\beta = 0.14$, $SE = 0.05$, $z(13184) = 2.50$, $p = .012$. Consequently, in the congruent group, the hit rate for the longest exposure duration of 2500 ms (M = 0.62, SD = 0.11) was significantly higher than for the shortest exposure duration of 1500 ms (M = 0.60, SD = 0.12), $t(102) = 2.36$, $p = .020$, $d_z = 0.23$, 95% CI

[0.04, 0.43]. For the incongruent group, we fit a generalized linear mixed model with a logit link function of the same structure as for the congruent group. The longer an image was presented during the learning phase, the more likely it became that participants in the incongruent group made a hit in a given target trial, $\beta = 0.15$, $SE = 0.06$, $z(13056) = 2.56$, $p = .010$ (see Supplementary 45). Consequently, in the incongruent group, the hit rate for the longest exposure duration of 2500 ms ($M = 0.60$, $SD = 0.16$) was significantly higher than for the shortest exposure duration of 1500 ms ($M = 0.57$, $SD = 0.14$), $t(101) = 2.60$, $p = .011$, $d_z = 0.26$, 95% CI [0.06, 0.45].

In an exploratory alternative analysis of the congruent group of experiment 2, we investigated whether the reward had an effect on whether participants responded "old" or "new". That is, we ran a drift diffusion model on all trials with the outcome variable "old" (lower boundary) or "new" (upper boundary). For the complex model, we let boundary separation, starting point, drift rate, and non-decision time vary by reward level (50, 750, 1450 or 2150 gems). Since no participants had to be excluded due to a lack of trials, the model was run on all $n = 103$ participants. When using the difference in BIC as criterion, the complex model was preferred in 100.00% of cases. When using the difference in AIC as criterion, the complex model was preferred in 100.00% of cases. According to the number of statistically significant likelihood ratio tests between the two models, the complex model provided a better fit in 45.63% of cases.

To estimate the effect of reward on boundary separation $\alpha$, we ran a linear mixed model of the form alpha ~ reward + (reward | participant) (see Supplementary Table 55). There was no statistically significant effect of reward on boundary separation, $\beta = 0.00$, $SE = 0.01$, $t(102.00) = 0.45$, $p = .656$ (see Supplementary Fig. 6a). That is, there was no statistically significant difference between boundary separation for the highest reward level ($M = 2.06$, $SD = 0.36$) and the lowest reward level ($M = 2.05$, $SD = 0.35$), $t(102) = 0.51$, $p = .608$, $d_z = 0.05$, 95% CI [−0.14, 0.24]. To estimate the effect of reward on the starting point $\beta$, we ran a linear mixed model of the form beta ~ reward + (reward | participant) (see Supplementary Table 56). The starting point decreased as the reward increased, $\beta = -0.01$, $SE = 0.00$, $t(102.00) = -3.14$, $p = .002$ (see Supplementary Fig. 6b). That is, the starting point for the highest reward level ($M = 0.48$, $SD = 0.08$) was significantly lower (closer to "old") than the starting point for the lowest reward level ($M = 0.51$, $SD = 0.08$), $t(102) = -3.24$, $p = .002$, $d_z = -0.32$, 95% CI [−0.52, −0.12]. A linear mixed model of the form delta ~ reward + (reward | participant) (see Supplementary Table 57) revealed a trend towards a decreasing drift rate as the reward increased, $\beta = -0.02$, $SE = 0.01$, $t(102.00) = -1.85$, $p = .068$ (see Supplementary Fig. 6c). However, there was no statistically significant difference between the drift rate for the highest reward level ($M = 0.06$, $SD = 0.23$) and the lowest reward level ($M = 0.10$, $SD = 0.23$), $t(102) = -1.63$, $p = .106$, $d_z = -0.16$, 95% CI [−0.35, 0.03]. A linear mixed model of the form tau ~ reward + (1 | participant) (see Supplementary Table 58) indicated a significant increase of the non-decision time as the reward increased, $\beta = 0.02$, $SE = 0.00$, $t(308.00) = 3.47$, $p<.001$ (see Supplementary Fig. 6d). That is, the non-decision time for the highest reward level ($M = 1.03$, $SD = 0.18$) was higher than for the lowest reward level ($M = 0.99$, $SD = 0.19$), $t(102) = 3.67$, $p<.001$, $d_z = 0.36$, 95% CI [0.16, 0.56]. In the congruent group of experiment 2, participants' median reaction times (in seconds) were slower when they made a decision for a high ($M = 1.90$, $SD = 0.47$) vs. a low reward ($M = 1.84$, $SD = 0.39$), $t(102) = 2.48$, $p = .015$, $d_z = 0.24$, 95% CI [0.05, 0.44].

For the incongruent group, we likewise ran an exploratory alternative analysis and investigated whether the shown reward had an effect on whether participants responded "old" or "new". That is, we ran a drift diffusion model on all trials with the outcome variable "old" (lower boundary) or "new" (upper boundary). For the complex model, we let boundary separation, starting point, drift rate, and non-decision time vary by reward level (50, 750, 1450 or 2150 gems). Since no participants had to be excluded due to a lack of trials, the model was run on all $n = 102$ participants. When using the difference in BIC as criterion, the complex model was

preferred in 100.00% of cases. When using the difference in AIC as criterion, the complex model was preferred in 100.00% of cases. According to the number of statistically significant likelihood ratio tests between the two models, the complex model provided a better fit in 60.78% of cases.

To estimate the effect of reward on boundary separation $\alpha$, we ran a linear mixed model of the form alpha ~ reward + (1 | participant) (see Supplementary Table 59). There was no statistically significant effect of reward on boundary separation, $\beta = 0.01$, $SE = 0.01$, $t(305.00) = 1.54$, $p = .124$ (see Supplementary Fig. 7a). That is, there was no statistically significant difference between boundary separation for the highest reward level ($M = 2.12$, $SD = 0.35$) and the lowest reward level ($M = 2.09$, $SD = 0.34$), $t(101) = 1.71$, $p = .090$, $d_z = 0.17$, 95% CI [−0.03, 0.36]. To estimate the effect of reward on the starting point $\beta$, we ran a linear mixed model of the form beta ~ reward + (reward | participant) (see Supplementary Table 60). The starting point decreased as the reward increased, $\beta = -0.02$, $SE = 0.00$, $t(101.00) = -3.87$, $p<.001$ (see Supplementary Fig. 7b). That is, the starting point for the highest reward level ($M = 0.48$, $SD = 0.09$) was significantly lower (closer to "old") than the starting point for the lowest reward level ($M = 0.52$, $SD = 0.08$), $t(101) = -3.76$, $p<.001$, $d_z = -0.37$, 95% CI [−0.57, −0.17]. A linear mixed model of the form delta ~ reward + (reward | participant) (see Supplementary Table 61) revealed that the drift rate decreased as the reward increased, $\beta = -0.04$, $SE = 0.01$, $t(100.99) = -3.54$, $p<.001$ (see Supplementary Fig. 7c). That is, the drift rate for the highest reward level ($M = 0.01$, $SD = 0.26$) was significantly lower than the drift rate for the lowest reward level ($M = 0.11$, $SD = 0.24$), $t(101) = -3.71$, $p<.001$, $d_z = -0.37$, 95% CI [−0.57, −0.17]. A linear mixed model of the form tau ~ reward + (reward | participant) (see Supplementary Table 62) revealed that the non-decision time increased as the reward increased, $\beta = 0.02$, $SE = 0.01$, $t(101.00) = 3.13$, $p = .002$ (see Supplementary Fig. 7d). That is, the non-decision time for the highest reward level ($M = 1.01$, $SD = 0.26$) was significantly higher than for the lowest reward level ($M = 0.98$, $SD = 0.26$), $t(101) = 2.82$, $p = .006$, $d_z = 0.28$, 95% CI [0.08, 0.48]. In the incongruent group of experiment 2, participants' median reaction times (in seconds) were slower when they made a decision for a high ($M = 1.90$, $SD = 0.49$) vs. a low reward ($M = 1.84$, $SD = 0.46$), $t(101) = 3.45$, $p<.001$, $d_z = 0.34$, 95% CI [0.14, 0.54].

### Experiment 3
The hit rate for the high reward level ($M = 0.64$, $SD = 0.14$) was significantly higher than for the low reward level ($M = 0.62$, $SD = 0.14$), $t(186) = 3.26$, $p = .001$, $d_z = 0.24$, 95% CI [0.09, 0.38] (see Fig. 7a).

The false alarm rate for the high reward level ($M = 0.34$, $SD = 0.14$) was significantly higher than for the low reward level ($M = 0.32$, $SD = 0.14$), $t(186) = 2.65$, $p = .009$, $d_z = 0.19$, 95% CI [0.05, 0.34] (Fig. 7b).

We preregistered a one-sided $t$-test to evaluate our hypothesis that memory strength ($d'$) would be higher for high than for low rewards. However, there was no statistically significant difference in $d'$ between the high ($M = 0.86$, $SD = 0.58$) and low reward ($M = 0.85$, $SD = 0.61$) in a preregistered, one-sided $t$-test, $t(186) = 0.42$, $p = .337$, $d_z = 0.03$, 95% CI [−0.09, $Inf$] (see Fig. 7c). ROC analyses corroborated the findings from the $d'$ analyses (see Supplementary Fig. 2). We preregistered that the confidence interval for the effect of reward on memory strength in experiment 3 would lie entirely above the effect size estimate in experiment 2 (congruent group), which was not the case. However, memory strength was significantly higher in experiment 3 ($M = 0.84$, $SD = 0.57$) than in experiment 2 ($M = 0.67$, $SD = 0.38$), $t(322.01) = 3.43$, $p<.001$, $d = 0.35$, 95% CI [0.15, 0.55].

Criterion was significantly more lenient for the high ($M = 0.04$, $SD = 0.32$) than for the low reward level ($M = 0.04$, $SD = 0.32$), $t(186) = -3.84$, $p<.001$, $d_z = -0.28$, 95% CI [−0.43, −0.13] (see Fig. 7d). That is, participants were more willing to identify a picture as old when the reward was high.

There was no statistically significant difference between the hit rate for the long duration level ($M = 0.63$, $SD = 0.13$) and the short duration level

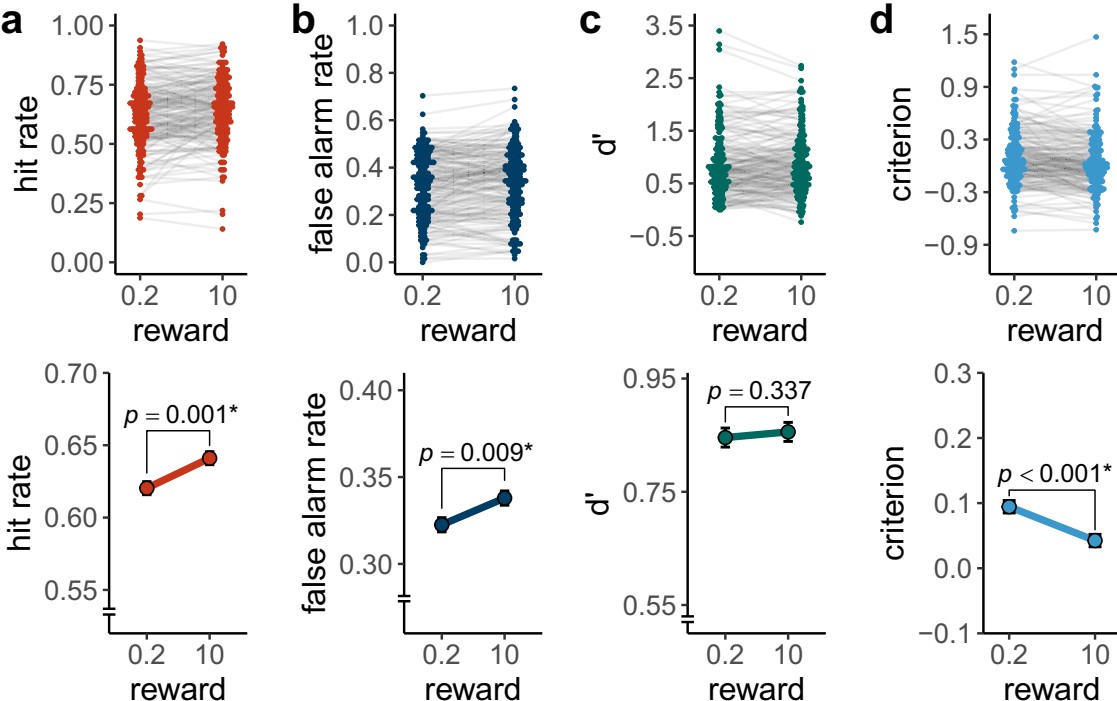

**Fig. 7 | Memory performance in experiment 3. a** Hit rate, **b** false alarm rate, **c** memory strength and **d** decision criterion for both reward based on $N = 187$ particpants. In the beeswarm plot of each panel, coloured dots show individual data points. Light grey lines connect the dots belonging to the same participant. The line plots of each panel show the group mean for each reward level. Black error bars show the within-subject standard error, as implemented in the R package Rmisc[50,51]. *P*-values for separate *t*-tests for hit rate, false alarm rate, memory strength and decision criterion are reported (see main text). Asterisks represent significance at $\alpha = .05$.

($M = 0.64$, $SD = 0.14$), $t(186) = -1.49$, $p = .137$, $d_z = -0.11$, 95% CI $[-0.25, 0.03]$.

In an exploratory analysis, we evaluated the influence of confidence and reward on the hit rate and the false alarm rate. We ran the following generalized linear mixed model with a logit link function on the target trials: hit (0 or 1) ~ confidence * reward + (confidence + reward | participant) + (confidence || image) (see Supplementary Table 46). In this model, we confirmed that the higher the reward, the more hits participants achieved, $\beta = 0.11$, $SE = 0.05$, $z(23936) = 2.43$, $p = .015$. Furthermore, participants had more hits with increasing confidence, $\beta = 0.92$, $SE = 0.07$, $z(23936) = 13.12$, $p<.001$. There was no statistically significant interaction between reward and confidence, $\beta = -0.01$, $SE = 0.04$, $z(23936) = -0.24$, $p = .810$. For lure trials, we ran the following generalized linear mixed model with a logit link function: false alarm (0 or 1) ~ confidence * reward + (confidence + reward || participant) + (confidence || image) (see Supplementary Table 47). For high rewards, participants made more false alarms, $\beta = 0.10$, $SE = 0.05$, $z(23936) = 2.25$, $p = .024$. Likewise, participants made more false alarms with increasing confidence, $\beta = -0.39$, $SE = 0.08$, $z(23936) = -4.84$, $p<.001$. There was no statistically significant interaction between reward and confidence, $\beta = -0.03$, $SE = 0.05$, $z(23936) = -0.63$, $p = .527$.

In an exploratory alternative analysis, we investigated whether the reward had an effect on whether participants responded "old" or "new". That is, we ran a drift diffusion model on all trials with the outcome variable "old" (lower boundary) or "new" (upper boundary). For the complex model, we let boundary separation, starting point, drift rate, and non-decision time vary by reward level (£ 0.2 or 10). Since no participants had to be excluded due to a lack of trials, the model was run on all $N = 187$ participants. When using the difference in BIC as criterion, the complex model was preferred in 99.47% of cases. When using the difference in AIC as criterion, the complex model was preferred in 99.47% of cases. According to the number of statistically significant likelihood ratio tests between the two models, the complex model provided a better fit in 39.04% of cases.

There was no statistically significant difference between boundary separation for the high reward level ($M = 2.15$, $SD = 0.35$) and the low reward level ($M = 2.15$, $SD = 0.35$), $t(186) = -0.04$, $p = .970$, $d_z = 0.00$, 95% CI $[-0.15, 0.14]$ (see Supplementary Fig. 8a). There was no statistically significant difference between the starting point for the high reward level ($M = 0.51$, $SD = 0.07$) and the low reward level ($M = 0.50$, $SD = 0.06$), $t(186) = -1.18$, $p = .238$, $d_z = -0.09$, 95% CI $[-0.23, 0.06]$ (see Supplementary Fig. 8b). The drift rate for the high reward level ($M = 0.00$, $SD = 0.22$) was significantly lower than the drift rate for the low reward level ($M = 0.05$, $SD = 0.21$), $t(186) = 3.05$, $p = .003$, $d_z = 0.22$, 95% CI $[0.08, 0.37]$ (see Supplementary Fig. 8c). There was no statistically significant difference between the non-decision time for the high reward level ($M = 0.83$, $SD = 0.33$) and the low reward level ($M = 0.82$, $SD = 0.31$), $t(186) = -0.49$, $p = .625$, $d_z = -0.04$, 95% CI $[-0.18, 0.11]$ (see Supplementary Fig. 8d). In experiment 3, there was no statistically significant difference in median reaction times between high ($M = 1.76$, $SD = 0.40$) and low reward decisions ($M = 1.75$, $SD = 0.40$), $t(186) = 1.24$, $p = .216$, $d_z = 0.09$, 95% CI $[-0.05, 0.23]$.

## Discussion

In three experiments, we investigated whether reward impacts memory strength at encoding or decision-making strategies at retrieval for reward-related long-term memory. We replicated the well-documented finding that reward increases the hit rate. While experiment 1 did not yield an interpretable false alarm rate per reward level, experiments 2 and 3 provided evidence that reward modulates decision-making strategies at retrieval. In the congruent group of experiment 2, we found an effect of reward on memory strength, but in experiment 3, there was no statistically significant effect of reward on memory strength. Crucially, in experiment 1, we found that participants displayed explicit knowledge of reward contingencies, which was related to the effect of reward on hit rate.

In experiment 1, we aimed to disentangle the contribution of memory strength versus decision-making strategies to this effect. For this,

participants rated their expected reward in order to calculate a false reward rate per reward level. However, reward expectations were highly biased towards medium rewards for false alarms. This prevented us from calculating a meaningful d', because it is unclear what the appropriate denominator for the false alarm rate would be. There are two alternatives: One is to divide the number of false alarms per expected reward level by the number of total trials per expected reward level (per participant). This is technically not a false alarm rate, because it pits false alarms against hits. The false alarm rate is normally calculated as the number of false alarms divided by the number of all lure trials - but since correct rejections are not associated with a reward value, this does not work on a reward-level basis. The other issue with this method is that the information that each false alarm rate per expected reward level is based on a different number of trials is lost. This might mask criterion effects, e.g., when participants are less likely to identify a picture as old when their reward expectations are lower, leading to less trials for the lowest expected reward category. The other alternative would be to divide the false alarms per expected reward level by a constant (e.g., 128, the total number of target trials - dividing by 32 trials, the number of trials per expected reward if lures were evenly distributed across rewards, does not work, because some participants would end up with a false alarm rate > 1). This way, a participant who, e.g., is heavily biased towards expecting medium reward levels will have a lower false alarm rate when expecting a high reward, which does not necessarily reflect higher accuracy or a more conservative response criterion. Lastly, when combining the hit rate and the false alarm rate into d', for both methods, the reward-dependent distribution of targets (uniform) and lures (more medium rewards) would differ, further complicating the interpretation of d'. These issues would be fixed if we had information about reward expectations for misses and especially correct rejections. However, we did not want to frustrate participants by asking them for their reward expectations when they just told us that they think a picture is new - and thus has no associated reward. Because of these issues, we concluded that the false alarm rate per expected reward level cannot be used to calculate a meaningful d'. Thus, we modified the paradigm again in experiment 2. For interested readers, we report d' analyses in the Supplementary Materials (both based on a denominator per expected reward level, and a fixed denominator; see Supplementary Note 6).

In our experiment 2, the influence of reward on false alarms differed between the congruent and incongruent group. In the congruent group, higher rewards increased the false alarm rate to a lesser degree than in the incongruent group – in fact, there was a decrease in false alarms between the third (1450 gems) and the fourth (2150 gems) reward level. Given that the reward-related increase in memory strength for the congruent group only emerged for the highest reward level, it is plausible that it is the reduction in false alarms for the highest reward level that is contributing to the effect. That is, both the hit rate and the false alarm rate increase as a function of reward for lower reward levels, reflected in a more lenient decision criterion. However, the reduction in false alarms for the highest reward level results in an increased memory strength for the highest reward level. Since the reward for lures is only presented during the test phase, it is not based on memory, however, the decrease in false alarms for the highest reward category might still be a result of memory processes. Participants in the congruent group could rely on their source memory especially for high reward targets. When encountering a new picture with a high reward, this might have resulted in a greater conflict with their source memory, which made them more careful. Participants in the incongruent group may have been surprised by the incongruence of their memory for stimulus-reward contingencies, and unable to use such strategies.

At first glance, it may be disappointing that hit rate reflects an influence of reward on decision-making strategies during retrieval rather than memory strength. However, for reward to influence decision-making processes in a memory test where no reward information is presented, participants need to remember which reward belongs to which picture, as was the case in our experiment 1. That is, participants must learn stimulus-reward associations, which are then used to make a decision at test. Along this line, it was recently shown that even memory for unchosen rewards can bias decision making[33].

Thus, pure measures of memory strength (i.e., d') do not isolate the processes underlying the memory-related modulation of performance by reward in the Motivated Learning Task. Instead, measures of the decision criterion and ideally, reward association memory, are preferred. However, modifying the Motivated Learning Task in such a way that these measures can be assessed is not trivial, and alternative versions of the task proposed so far all come with drawbacks. This is why we conclude that the standard version of the task, with the hit rate as outcome measure, might be a valid option in many scenarios. However, it should not be interpreted as a measure of memory strength, but rather as a measure of a memory-based criterion.

Our third experiment with only two reward categories and perfect information about the rewards at recognition is most comparable to prior work that – as we did in experiment 3 – finds that reward does not influence memory strength, but only decision criterion. Two interpretations are possible: There might be an effect of reward on memory strength, but it could be so small that it cannot be detected reliably with even large sample sizes as in our study (around $N = 100$), let alone with "standard" sample sizes in the field (around $N = 30$). This would make it very hard to assess these effects in more resource-heavy neuroscientific experiments. The alternative explanation is that effects of reward on memory strength only arise under very specific conditions. For example, the reward-based change in memory strength in our second experiment was largely driven by the highest reward level, while the lower reward levels did not differ in the effect on memory strength. It might be the case that effects of reward on memory strength only emerge when there are at least more than two reward levels, allowing participants to focus on the comparably few trials with a very high reward, as compared to half of the trials with a high reward in the case of two reward levels. This potential moderator of reward context should be tested in future research.

Reward contingencies are a form of source memory (i.e., participants remember the reward associated with a stimulus). In support of this interpretation, we found participants to have more accurate reward expectations when their confidence was high. We did not ask for remember/know judgements, to not further burden our participants, and can thus not distinguish remember-based responses from familiarity-based responses[34]. Similar to our findings, Adcock et al.[2] report that – unlike for high-confidence responses – there was no statistically significant effect of reward on low-confidence responses. Likewise, Shigemune et al.[4] found that rewards modified item-source associations, but did not affect item memory alone. Our observation that more accurate memory of stimulus-reward associations was correlated with a reward-related increase in hit rate provides further evidence that the effects of reward on hit rate rely on source memory.

Bowen et al.[20] associated categories with different amounts of reward to give lures an objective value. Their approach provides perfect information of reward contingencies and removes the need to form stimulus-reward associations, since participants rely on perfectly learned category-reward associations. This decouples decision-making during retrieval from memory processes. We tried to design a paradigm without perfect reward information that would still yield a false alarm rate per reward level in experiment 1, but were not successful. In experiment 2 and 3, we presented rewards during the test phase, providing an objective reward-modulated false alarm rate. As the paradigm used by Bowen et al.[20], this means that participants had (or believed to have) perfect information about reward contingencies, so could base decision on cues presented during retrieval rather than memory for stimulus-reward associations. However, we directly manipulated the reward information presented to the participants, which was either accurate (congruent group) or not (incongruent group). The differences between the congruent and incongruent group provide empirical evidence that reward information presented during retrieval influences participants' behaviour. Crucially, our set of studies demonstrates that the reward associations that bias decision making during retrieval are part of the construct that is of interest to researchers studying long-term memory for rewards rather than an unfortunate side effect of the measurement.

It is possible that providing participants with reward information during recognition changes the cognitive processes participants use to complete the task. I.e., when participants are not presented with any reward information during recognition (experiment 1), they might rely more strongly on (stimulus-reward) memory. When reward information is provided during recognition (experiments 2 and 3), participants might shift their criterion based on the presented reward, and disregard previously formed (stimulus-reward) memory. While we cannot rule out this explanation entirely, we argue against it: Overall memory performance is descriptively even better in experiments 2 and 3 than in experiment 1 (see Supplementary Table 4), and better performance can only be achieved by improved discrimination between old and new pictures. Furthermore, reward information presented during recognition is not neutral, but seems to influence memory performance: For participants in the incongruent group, memory strength is worse when they are more strongly influenced by the shown reward. In line with this, response patterns in experiment 2 differ between the congruent and the incongruent group, especially for false alarms. This suggests that even when reward information is presented during recognition, knowledge about stimulus-reward associations is used to retrieve matching memory traces. If the reward information presented during recognition would override the participants' memory entirely, there would be no difference in performance between the congruent and incongruent group. We argue that rather than overriding existing memory traces, the reward information presented during recognition is used as a memory cue that guides retrieval. Lastly, it would be maladaptive if participants disregarded their memory of the pictures in favour of the reward information presented during recognition.

Generally, it makes sense that animals form stimulus-reward associations, rather than only relying on strengthening of the memory trace for the highly rewarded stimulus. Remembering a stimulus strongly without information about its reward contingencies prevents the organism from adapting to changing reward contexts. For example, scrub jays memorize where they cached their highly preferred food, mealworms, but also remember the location of less preferred food, peanuts[35]. Restricting access to the mealworms until they have gone bad, leads to the birds reliably retrieving less perishable peanuts. Likewise, participants in the Motivated Learning Task use stimulus-reward information encoded during learning or provided at retrieval testing to optimize decision-making during retrieval. This interpretation aligns with findings that the effects of reward on decision-making are sensitive to modulations of punishments for incorrect decisions[20], and that rewards influence retrieval strategies[36]. This account also connects the present research to research about model-based versus model-free learning in reinforcement learning tasks[37]. For model-based behaviour the participant can use information about expected stimulus-reward contingencies to adapt the criterion and maximize monetary gain. Flexible model-based representations enable participants to generalize knowledge about rewards to new stimuli across different contexts[38]. This is in line with recent evidence that dopamine signals causal associations between cue and reward[39]. Model-based explanations oppose accounts of more automatic influences of reward on memory, e.g., automatic dopaminergic processes during encoding strengthening the subsequently formed memory trace[2]. In tasks where participants have a good, explicit representation of the task structure (which we ensured with our validation questions about the reward contingencies), behaviour is mainly model-based[40,41]. In contrast, when learning is incidental, prediction errors based on reward probability influence (automatic) strengthening of the memory trace[42]. Future research could relate inter-individual differences in reward-associated memory performance to model-based versus model-free decision-making tendencies.

Our experiments reach similar conclusions as the experiments by Bowen et al.[20]. However, our study goes beyond them since in contrast to their work, we are able to examine the effect of associative reward memory and the item-specific effects of showing rewards at retrieval, which the authors draw implicit conclusions about without directly investigating them. Detailing on this, we test participants' source memory for the rewards in experiment 1, which provided crucial insights into participants' reward expectations. Like Bowen et al.[20], our version of the task in experiments 2 and 3 provided participants with perfect reward information, minimizing the need for source memory during recognition. What makes our version of the task different is that we were able to manipulate the congruency of the reward information presented during the test phase. We believe that our results complement and go beyond the results of Bowen et al.[20]. In addition, our large sample allows for a more precise and more robust estimate of effect sizes. We demonstrate that reward effects on memory can be categorized as small[43], which means that common sample sizes around 30 participants are not enough to detect them reliably and therefore the work by Bowen et al.[20] was likely not powerful enough to find subtle effects on sensitivity and thus their conclusions based on absence of evidence were premature.

While our conclusions are based on our preregistered mixed model analyses, it is helpful to look at the data from a different perspective. Our alternative analysis based on drift diffusion models, which take reaction times into account, generally confirms our findings that reward influences participants' performance at retrieval. In experiment 1, we found that reward expectations do not bias participants in the direction of a hit or false alarm (starting point). Note that the starting point cannot be interpreted as equivalent to a decision criterion in this model, because the outcome was "hit"/"false alarm", and not "old"/"new". That is, we cannot draw any conclusions about decision criteria with this model. The drift rate increased as reward expectations increased, which means evidence accumulation was faster and more accurate when participants expected a higher reward, potentially due to more noisy memory traces for low rewards. There was a trend towards a decreasing non-decision time as the reward increased; the non-decision time was higher for low rewards than for high rewards. That is, descriptively, participants needed more time for components that are not part of the decision process (e.g., early stimulus processing, preparing the motor response) when the reward was very low. Memory processes during early retrieval of stimuli might also contribute to the non-decision time. According to the REM (retrieving effectively from memory[44]) model, each item in memory is stored as a vector of features. During a recognition memory test, the probe vector of the test item is compared to the vectors in memory to determine whether the item is old or new. While the comparison of the vectors could be regarded as part of the evidence accumulation phase, earlier phases of the REM model, e.g., compiling an initial set of vectors that the probe vector is compared to, could affect the non-decision time. Within this framework, it might be more effortful to generate a set of possible matches for low-reward items, because their representation in memory is weaker.

For both the congruent and incongruent group of experiment 2, we found that the higher the reward, the more participants were biased towards responding "old", which corresponds to the effect of reward on the decision criterion we found in the main analysis. We found a reduced drift rate as the reward increased (a trend for the congruent group; a significant effect for the incongruent group). This may seem at odds with the findings of experiment 1, where the drift rate increased for higher rewards. However – in contrast to experiment 1 – in experiment 2, the reward information was presented before the test stimulus was revealed. Participants might spend additional effort on stimuli labelled as high rewards, even if they do not recognize them. This might make the evidence accumulation for stimuli labelled as high rewards more noisy, resulting in an increased drift rate. In experiment 1, on the other hand, participants might quickly skip over stimuli that do not have a strong representation in memory. A similar explanation could apply to the non-decision time, which also increased as the reward increased: Participants might spend more time on processing a stimulus following a high-reward cue during the test phase, or generating potentially matching vectors in memory. The findings of the drift diffusion models for experiment 3 follow a different pattern. There was no statistically significant association between reward and starting point, but the drift rate was significantly lower for the high reward level. It is intriguing that experiment 3 had a different pattern of results in both the main analysis and the exploratory alternative drift diffusion analysis. A speculative explanation could be that processing

two reward levels is different from processing four reward levels. With two reward categories, there are more stimuli per category – potentially too many to prioritize the high rewards.

Taken together, these findings suggest that participants spend more effort when a stimulus is announced as a high-reward item, regardless of whether its memory representation is strong. In contrast, decisions based on reward expectations require less effort, because resources are not "wasted" on stimuli that are only represented weakly in memory. Our results might further indicate that non-decision processes might be related to memory sensitivity. Lastly, it should be noted that because they also take reaction times into account, the parameters of drift diffusion models do not directly correspond to signal detection measures such as memory strength and decision bias. Thus, the drift diffusion models reported here provide complementary information to the analyses in the main manuscript, rather than challenging their interpretations.

In previous literature investigating the influence of reward on decisions and reaction times, the phenomenon of "magnitude sensitivity" has been described: "a choice between equal alternatives of high magnitude is made faster compared with a choice between equal alternatives of low magnitude", where "magnitude" means the summed value of the alternatives[45]. In our experiments, we find the opposite pattern: Participants generally make slower decisions for stimuli preceded by a high reward. The most striking differences between our paradigm and those reported in the literature and magnitude sensitivity is that our participants a) do not choose between multiple alternatives presented on the screen and b) make their decision based on memory. To our knowledge, there are no studies investigating the classical magnitude sensitivity effect in memory-based paradigms. Even though our Motivated Learning Task differs in several aspects from typical magnitude sensitivity paradigms, our data might provide tentative evidence that reward magnitude affects memory-based decisions differently.

## Limitations

While experiment 1 enabled us to investigate participants' reward expectations and experiments 2 and 3 allowed us to calculate a reward-dependent false alarm rate, we were not able to create a paradigm that solved both problems at the same time. Future research would greatly benefit from tasks that do not need to provide their participants with perfect information about rewards, but still allow researchers to calculate a reward-dependent false alarm rate. A step in this direction might be an elegant two-step paradigm by Jang et al.[42], who associated reward probabilities with image categories, which changed over the course of the experiment. This provided them with a memory-based false alarm rate per reward probability. They found that reward prediction errors based on reward probability were associated with memory strength, but did unfortunately not report a measure of criterion.

Furthermore, in experiment 1, reward expectations were only available for "old" responses, which means that for correct rejections or misses, information about reward expectations is missing, limiting our analyses. We decided not to ask participants about the reward they expected to receive for their "new" decisions because that might have confused or frustrated participants. After all, when they identify a picture as new, they do not expect that the picture is associated with a reward.

In experiment 1 and 2, we used four different reward levels to investigate potential effects of reward in a more fine-grained manner, which might make our findings less comparable to previous literature, where it is common to use only two reward levels[2,8,20]. It is possible that rewards are processed differently when there are more reward levels. For example, since there are more images per reward level, participants might have less associative memories for each reward level. Less stimuli per reward level might allow them to prioritize the most salient information (i.e., the highest reward level). This would explain why results for experiment 3, where we used only two reward levels, differed from those for the congruent group of experiment 2, even though we cannot attribute the different results between the two experiments to the difference in reward levels with complete certainty. Future work should systematically investigate the influence of the number of reward levels on memory performance. However, another methodological limitation for paradigms that employ more reward levels is that less information (number of trials) per reward level is available, limiting statistical analyses.

Another limitation of our (and many previous) studies is the comparably small amount of money that participants could earn during the task. However, the striatal response to rewarding stimuli is not only modulated by absolute magnitude but also by relative magnitude of rewards[46–48]. In addition, our use of gems as rewards instead of money may be criticized. However, in classical tasks such as the Monetary Incentive Delay task participants are able to transfer reinforcement properties to novel reward cues fast[27,49]. Additionally, we found no relative advantage of using monetary cues instead of gems in experiment 3.

## Conclusion

In conclusion, the highly robust effect of reward on hit rate reflects decision criteria at retrieval rather than changes in memory strength during encoding or consolidation. We propose that the decision criterion at recognition is based on memory for stimulus-reward associations learned during encoding and consolidated thereafter. However, it will be important to develop tasks that can reliably measure memory for stimulus-reward associations and reward effects on memory strength within one task in the future.

## Data availability

All data (https://doi.org/10.23668/psycharchives.13963) are publicly available at the PsychArchives of the Leibniz Institute of Psychology (ZPID).

## Code availability

All analysis code (https://doi.org/10.23668/psycharchives.13967) is publicly available at the PsychArchives of the Leibniz Institute of Psychology (ZPID). All analyses and plots were generated using R (version 4.3.1 (2023-06-16 ucrt)[52]. Effect sizes were calculated using the R package effectsize (version 0.8.5[53]). (Generalized) linear mixed models were calculated using the R package lme4 (version 1.1.34[29]), and $p$-values for these models were calculated using the R package lmerTest (version 3.1.3[54]). Drift diffusion models (DDM) were fit using the wdm() function from the R-package RWiener (version 1.3.3[55]). Plots were created using the following packages: ggplot2 (version 3.4.3[56]), ggbeeswarm (version 0.7.2[57]), cowplot (version 1.1.1[58]), ggpubr (version 0.6.0[59]), ggh4x (version 0.2.5[60]), and scales (version 1.2.1[61]).

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

## Acknowledgements

This research was supported by an Emmy-Noether-Grant (FE 1617/2-1) to Gordon B. Feld. Simon Kern was supported by a PhD scholarship by the Studienstiftung des Deutschen Volkes. The funding sources had no role in the study design, or in the collection, analysis and interpretation of data, in the writing of the report, and in the decision to submit the article for publication. We would like to thank Eduard Ort for his advice on the drift diffusion models.

## Author contributions

Juliane Nagel: developed the study concept and study design, generated the study material, performed data collection, performed the data analysis and interpretation, drafted and revised the manuscript. David Philip Morgan: developed the study concept and study design, generated the study material, performed data collection, drafted the manuscript. N. Çağatay Gürsoy: Provided critical feedback on the study design, generated the study material. Samuel Sander: performed data analysis and interpretation. Simon Kern: provided critical feedback on the study design. Gordon B. Feld: developed the study concept and design, supervised data analysis and interpretation, revised the manuscript. All authors approved the final version of the manuscript for submission.

## Competing interests

The authors declare no competing interests.
