## [Peer Review File · Communications Psychology]

10th Jul 23

Dear Ms Nagel,

Thank you for your patience during the peer-review process. Your manuscript titled "Memory for Rewards Guides Retrieval" has now been seen by 3 reviewers, whose comments are appended below. You will see that they find your work of some potential interest. However, they have raised quite substantial concerns that must be addressed. In light of these comments, we cannot accept the manuscript for publication, but would be interested in considering a revised version that fully addresses these serious concerns.

We hope you will find the Reviewers' comments useful as you decide how to proceed. Should additional work allow you to address these criticisms, we would be happy to look at a substantially revised manuscript. If you choose to take up this option, please highlight all changes in the manuscript text file, and provide a detailed point-by-point reply to the reviewers.

Editorially, we consider it key that your revised version comprehensively addresses the referees' methodological concerns. Although the reviewers primarily highlight opportunities for additional analyses that would strengthen your interpretation, note that the revision may require additional data collection where the amount of existing data prevents robust analysis. Ambiguities that remain despite additional empirical work must be transparently discussed, including in a section titled "Limitations" that forms part of the Discussion.

As you revise the manuscript, please complete the checklist linked below to ensure that your revised work aligns with our editorial and formatting requirements.

If the revision process takes significantly longer than five months, we will be happy to reconsider your paper at a later date, provided it still presents a significant contribution to the literature at that stage.

Please use the following link to submit your revised manuscript, point-by-point response to the Reviewers' comments with a list of your changes to the manuscript text (which should be in a separate document to any cover letter) and any completed checklist:

[link redacted]

** This url links to your confidential home page and associated information about manuscripts you may have submitted or be reviewing for us. If you wish to forward this email to co-authors, please

delete the link to your homepage first **

Please do not hesitate to contact me if you have any questions or would like to discuss the required revisions further. Thank you for the opportunity to review your work.

Best regards,

Marike

Marike Schiffer, PhD
Chief Editor
Communications Psychology

EDITORIAL POLICIES AND FORMATTING

Editorial Policy: Policy requirements (Download the link to your computer as a PDF.)

Furthermore, please align your manuscript with our format requirements, which are summarized on the following checklist:

Communications Psychology formatting checklist

and also in our style and formatting guide Communications Psychology formatting guide .

* **CODE AVAILABILITY:** All Communications Psychology manuscripts must include a section titled "Code Availability" at the end of the methods section. In the event of publication, we require that the custom analysis code supporting your conclusions is made available in a publicly accessible repository; please choose a repository that provides a DOI for the code; the link to the repository and the DOI must be included in the Code Availability statement. Publication as Supplementary Information will not suffice. We ask you to prepare and upload code at this stage, to avoid delays later on in the process.

* **DATA AVAILABILITY:**

All Communications Psychology research manuscripts must include a section titled "Data

Availability" at the end of the Methods section or main text (if no Methods). More information on this policy, is available at <http://www.nature.com/authors/policies/data/data-availability-statements-data-citations.pdf>.

At a minimum the Data availability statement must explain how the data can be obtained and whether there are any restrictions on data sharing. Communications Psychology strongly endorses open sharing of data. If you do make your data openly available, please include in the statement:

We recommend submitting the data to discipline-specific, community-recognized repositories, where possible and a list of recommended repositories is provided at <http://www.nature.com/sdata/policies/repositories>.

If a community resource is unavailable, data can be submitted to generalist repositories such as figshare or Dryad Digital Repository. Please provide a unique identifier for the data (for example a DOI or a permanent URL) in the data availability statement, if possible. If the repository does not provide identifiers, we encourage authors to supply the search terms that will return the data. For data that have been obtained from publicly available sources, please provide a URL and the specific data product name in the data availability statement. Data with a DOI should be further cited in the methods reference section.

REVIEWER EXPERTISE:

Reviewer #1: cognitive science, memory

Reviewer #2: cognitive science, memory

Reviewer #3: cognitive science, decision making

Reviewer #1 (Remarks to the Author):

Nagel and colleagues report the results of several experiments focused on the role of reward in enhancing episodic memory. Previous work in this area has shown that when items are associated with a higher reward at encoding, participants are more likely to remember those items later. However, most of this work has failed to distinguish between a change in memory strength and a change in decision-making criterion. Across three large online experiments, the authors attempt to disambiguate these two explanations. Consistent with prior work, the authors found that items associated with a higher reward at encoding have a higher hit rate during a later recognition memory test. By assigning reward values to lure items, the authors also show that higher rewards make people more likely to false alarm. Thus, across multiple experiments, the authors demonstrate that higher rewards cause people to use a more lenient criterion at retrieval: they are more likely to endorse *both* old and new items as old. The authors report weak and inconsistent effects of reward on d' , pushing back on the narrative that reward increases memory strength at the time of

encoding. Instead, their results point toward an effect of reward on decision-making strategy at the time of retrieval.

This paper addresses an interesting and important question about how reward changes memory expression. The manipulations performed here add value to the literature and the experiments are well powered. I have some questions and concerns about the analyses and conclusions, included below.

Major Comments:

1. The authors decline to compute false alarm rate, d' , and criterion from their data from Experiment 1 because of an unequal number of observations across expected reward levels for lures. They demonstrate statistically that the observations are unequally distributed, but I'm unconvinced based on their paper that they can't analyze this data. It's common in experimental psychology to have uneven observations of confidence, remembered/forgotten, etc. Even if some participants don't have any observations at a particular expected reward level, their mixed model approach should be able to handle this. More clarity is needed about the author's decision here.
2. The authors report effects of duration and confidence in Experiment 1 but don't report effects of duration in Exp 2 or 3 or confidence in Experiment 3. It would be good to report these results for completeness (if only in the text). The authors should also consider adding stimulus duration to their mixed models that evaluate reward since it will account for some variance and this variable could interact with reward. If there is some other reason stimulus duration was manipulated it should be stated; it's unclear from the paper why this was done.
3. In experiment 2, the effect of reward on memory strength (d') for congruent trials looks very nonlinear across reward values (no change across the first three reward levels and a big change in the final reward level). The authors report a significant linear effect of reward value on d' , but this clearly doesn't describe the effect. Is this effect still significant if the authors allow reward to vary nonlinearly or code it categorically? Given that this is the only significant effect of reward on memory strength reported in the paper, it would be good to ensure robustness. It may also be worth discussing what's driving the effect – there's a *reduction* in false alarm rate from the second highest to highest reward level but an increase in the hit rate. How do the authors interpret this?
4. I thought that participants with chance level or below chance level d' were excluded, but there are some negative d' values in Figure 5A and 5B. Can the authors clarify?
5. I'm confused by the conclusion reached in the abstract and discussion that this paper validates the use of hit rate to study this reward-related phenomenon, e.g. "In this framework, the hit rate, that includes information about decision-making processes, better reflects the processes of interest than pure measures of memory strength such as the sensitivity index". The authors clearly identify the problem with using hit rate; it reflects both memory strength and decision-making processes and they can't be dissociated. Future studies relying on hit rate will also not be able to interpret results in favor of one or the other. It seems to me that criterion best isolates the process underlying the behavioral effect of interest.
6. In the final paragraph of the discussion the authors state, "We propose that hit rate is a valid

measure of associative memory for stimulus-reward pairs learned during encoding and consolidated thereafter.” It is interesting and compelling that the authors show that the effect of reward on hit rate is related to participants’ ability to correctly recall/predict the reward associated with each stimulus (Experiment 1). However, the effect size of this correlation is modest and it seems strange to claim that item recognition memory is a valid measure of associative memory, when we can test associative memory explicitly instead. The authors should consider rephrasing to avoid saying this.

7. The authors could be more clear about how their paper goes beyond the Bowen 2020 paper, which reaches a similar conclusion.

Minor Comments:

1. The paper would be more clear if the task figure came first, rather than last.

2. I’m not familiar with the notation “reward || participant” in the lmer models. Is this distinct from “reward | participant”? If so, the authors should clarify and justify when one is used versus the other.

Reviewer #2 (Remarks to the Author):

The main goal of this set of three experiments was to examine whether reward modulates memory strength or decision biases. Memory for stimuli encoded with varying amount of reward was measured with metrics of hit rate, false alarm rate, confidence, and in some experiments memory sensitivity and criterion. In experiment 1, reward value modulated hit rate, and confidence, but false alarm could not be calculated at each level of reward. In Experiment 2, rewards were presented on the screen at encoding and at retrieval, and both memory sensitivity and decision biases were affected by reward value. In Experiment 3, changes to the paradigm were made that resulted in effects of reward on hit rate, false alarm rate and criterion, but not memory sensitivity. This is an interesting and creative study replicating and extending some prior work to better understand the mechanisms associated with motivated memory. A strength of the studies are the large sample sizes. I have some feedback about the interpretation of the findings as well as some minor comments.

As noted in the introduction, one of the main goals of the study is to create a paradigm where memory sensitivity and decision making can be separated without presenting “perfect information” about the reward information at retrieval. While no reward information was presented at test in Exp. 1, these metrics could not be calculated. In the other two experiments, these metrics were calculated, but reward information was also presented at test. It seems one of the main aims of the study to really test these research questions about mechanism was not really achieved. Please provide more discussion of this in the paper.

Relatedly, one main take-away -- hit rates contain essential information about learning, and how d' doesn't isolate memory processes-- needs more clarification (lines 453-458). It isn't clear to me from the results why this is the conclusion from study 1 given that the false alarm rate information was not usable. Further, Experiment 3 is most similar to the studies that the authors are comparing to, and the findings replicate prior work with no effect on memory sensitivity but effects on response bias, but this is glossed over in the general discussion.

It seems to me that the main conclusion from the results is that when no information about reward is explicitly presented at retrieval participants may rely more on memory, but when information is presented at retrieval, this overrides memory strength and participants rely on decision biases. This is touched on by the authors but is sort of buried with the discussion.

The rationale for different exposure durations during encoding, and why analysis of this only occurred in Experiment 1, is also not clear.

Reviewer #3 (Remarks to the Author):

In their paper, Nagel and colleagues present a series of 3 online experiments to better understand whether anticipated rewards have a direct impact on memory strength or an indirect via changing the decision strategy at retrieval. They use the Motivated Learning Task with new modifications, the first being to ask participants about their expected reward, the second being to remind participants about the associated reward. They replicate the (positive) effect of reward on recall success, but the first experiment's results are questionable (according to the authors' own statement), because the reported expected rewards were biased. In the second and third experiment, they find evidence for a change of decision strategy at retrieval.

Overall, the paper addresses an interesting topic, the interplay of memory, reward and decision making, and it is (mostly) well written and good to understand. I appreciate the authors' rigorousness w.r.t. good scientific practices (preregistration, data and code availability to reviewers, honesty about their goals and predictions, and critical evaluation of their own work). All of this being said, I am still not convinced that this study provides a substantial contribution to the existing literature. The first experiment is difficult to interpret, as the authors acknowledge themselves. The second experiment seems to support the "strategy" hypothesis. This seems to be a conceptual replication of Bowen et al. (2020), to which the authors refer a lot, which is okay. However, the authors appear to be a bit reluctant to accept this view, which is probably why they took out the incongruent condition in experiment 3 and tried to make the task easier. This, however, does not turn out to change the general results. So, overall, they made many, perhaps too many, unfortunate choices regarding the experimental design, and the result of all of this is that we have (at best) a conceptual replication of another study. Also, I was a bit surprised that they would not look deeper into the "strategy" hypothesis and analyze response time (RT) data.

Further comments:

1. As just stated, I was surprised that RT data were not analyzed. If I understand the authors correctly, the idea of the "strategy" hypothesis is that, if participants recall the associated (larger) reward of an old item at retrieval, they "try harder" to be correct to get the larger reward. This strategy adaptation would in my view imply that participants take more time to make their decision, in the sense of an increased threshold / boundary separation in a sequential sampling model / drift-diffusion model (DDM). It would be interesting to see whether there is evidence for this effect. Because reward could have additional effects (including a change in sensitivity), the best approach would obviously be to apply these models to the data. In my view, a change in sensitivity (similar to d' in SDT) would be related to a change in drift rate, whereas a change in strategy (similar to

criterion in SDT) would be related to a change in the threshold and/or starting point. The advantage of SSM/DDM over SDT is, that the RT data might contain very critical information to answer this question.

2. In the sub-section "Expected Reward" of the results of the first experiment, an analysis of confidence rating is reported, but it has never been mentioned before that confidence ratings were obtained. This should be mentioned earlier (when explaining the task design), and the rationale for including (and analyzing) them should be given. Relatedly, in the same section the false alarm result of Figure 1B is reported, but this should be reported earlier on (the authors jump back and forth

3. "For participants whose reward expectations were more accurate, the hit rate was more strongly influenced by true reward, $r(198)=.24$, $p<.001$ (Figure 2C)." Is this still one of the preregistered hypotheses? Relatedly, the analysis for Figure 2D feels highly circular/redundant to the analysis for Figure 2C

4. "Since it is unclear what the false alarm rate based on the expected reward represents" Plotting the false alarm rate as a function of expected reward would be informative.

Signed,
Sebastian Gluth

Any changes in the manuscript and the supplementary material have been highlighted in yellow. The relevant sections of the manuscript have also been pasted below the respective reviewer comment for convenience.

As requested in the journal formatting guidelines, limitations have now been added in a separate section:

“While experiment 1 enabled us to investigate participants’ reward expectations and experiments 2 and 3 allowed us to calculate a reward-dependent false alarm rate, we were not able to create a paradigm that solved both problems at the same time. Future research would greatly benefit from tasks that do not need to provide their participants with perfect information about rewards, but still allow researchers to calculate a reward-dependent false alarm rate. A step in this direction might be an elegant two-step paradigm by Jang et al. (2019), who associated reward probabilities with image categories, which changed over the course of the experiment. This provided them with a memory-based false alarm rate per reward probability. They found that reward prediction errors based on reward probability were associated with memory strength, but did unfortunately not report a measure of criterion.

Furthermore, in experiment 1, reward expectations were only available for “old” responses, which means that for correct rejections or misses information about reward expectations is missing, limiting our analyses. We decided not to ask participants about the reward they expected to receive for their “new” decisions because that might have confused or frustrated participants. After all, when they identify a picture as new, they do not expect that the picture is associated with a reward.

In experiment 1 and 2, we used four different reward levels to investigate potential effects of reward in a more fine-grained manner, which might make our findings less comparable to previous literature, where it is common to use only two reward levels (Adcock et al., 2006; Bowen et al., 2020; Feld et al., 2014). It is possible that rewards are processed differently when there are more reward levels. For example, since there are more images per reward level, participants might have less associative memories for each reward level. Less stimuli per reward level might allow them to prioritize the most salient information (i.e., the highest reward level). This would explain why results for experiment 3, where we used only two reward levels, differed from those for the congruent group of experiment 2, even though we cannot attribute the different results between the two experiments to the difference in reward levels with complete certainty. Future work should systematically investigate the influence of the number of reward levels on memory performance. However, another methodological limitation for paradigms that employ more reward levels is that less information (number of trials) per reward level is available, limiting statistical analyses.

Another limitation of our (and many previous) studies is the comparably small amount of money that participants could earn during the task. However, the striatal response to rewarding stimuli is not only modulated by absolute magnitude but also by relative magnitude of rewards (Cromwell et al., 2005; Tobler et al., 2005; Vaidya et al., 2013). In addition, our use of gems as rewards instead of money may be criticized. However, in classical tasks such as the Monetary Incentive Delay task participants are able to transfer reinforcement properties to novel reward cues fast (Kirsch et al., 2003; Knutson et al., 2000). Additionally, we found no relative advantage of using monetary cues instead of gems in experiment 3.”

Reviewer 1

Major Comments:

1. The authors decline to compute false alarm rate, d' , and criterion from their data from Experiment 1 because of an unequal number of observations across expected reward levels for lures. They demonstrate statistically that the observations are unequally distributed, but I'm unconvinced based on their paper that they can't analyze this data. It's common in experimental psychology to have uneven observations of confidence, remembered/forgotten, etc. Even if some participants don't have any observations at a particular expected reward level, their mixed model approach should be able to handle this. More clarity is needed about the author's decision here.

We now elaborate on the difficulties related to d' in the interim discussion of experiment 1:

“[...] However, reward expectations were highly biased towards medium rewards for false alarms. This prevents us from calculating a meaningful d' , because it is unclear what the appropriate denominator for the false alarm rate would be. There are two alternatives: One is to divide the number of false alarms per expected reward level by the number of total trials per expected reward level (per participant). This is technically not a false alarm rate, because it pits false alarms against hits. The false alarm rate is normally calculated as the number of false alarms divided by the number of all lure trials - but since correct rejections are not associated with a reward value, this does not work on a reward-level basis. The other issue with this method is that the information that each false alarm rate per expected reward level is based on a different number of trials is lost. This might mask criterion effects, e.g., when participants are less likely to identify a picture as old when their reward expectations are lower, leading to less trials for the lowest expected reward category. The other alternative would be to divide the false alarms per expected reward level by a constant (e.g., 128, the total number of target trials - dividing by 32 trials, the number of trials per expected reward if lures were evenly distributed across rewards, does not work, because some participants would end up with a false alarm rate > 1). This way, a participant who, e.g., is heavily biased towards expecting medium reward levels will have a lower false alarm rate when expecting a high reward, which does not necessarily reflect higher accuracy or a more conservative response criterion. Lastly, when combining the hit rate and the false alarm rate into d' , for both methods, the the reward-dependent distribution of targets (uniform) and

lures (more medium rewards) would differ, further complicating the interpretation of d' . These issues would be fixed if we had information about reward expectations for misses and especially correct rejections. However, we did not want to frustrate participants by asking them for their reward expectations when they just told us that they think a picture is new - and thus has no associated reward. Because of these issues, we concluded that the false alarm rate per expected reward level cannot be used to calculate a meaningful d' . Thus, we modified the paradigm again in experiment 2. For interested readers, we report d' analyses in the supplement (both based on a denominator per expected reward level, and a fixed denominator; see Supplementary Materials, section: "Experiment 1 d' and criterion")."

That is, the problem occurs at the level of the aggregated data (the false alarm rate), not at the level of the trial-wise data, where a mixed model would indeed be able to handle missing/unequal observations (but no d' can be calculated). For more insight into how the reward expectations influence the false alarms, we have analysed the data at trial level, and report our analyses in the new "False alarms" section of experiment 1 (also see the replaced/added Figures 2B and C):

"Since lures do not have a true reward, we evaluated the influence of the expected reward on false alarms. That is, for trials where participants said that an image was "old", we ran the following generalized linear mixed model with a logit link function: $\text{false alarm (0 or 1)} \sim \text{expected reward} * \text{confidence} + (\text{expected reward} \parallel \text{participant}) + (1 \parallel \text{image})$. With increasing confidence, participants made less false alarms, $\beta = -0.49$, $SE = 0.04$, $z(25711) = -13.81$, $p < .001$. There was no significant main effect of expected reward on false alarms, $\beta = 0.01$, $SE = 0.03$, $z(25711) = 0.31$, $p = .755$, but an interaction between expected reward and confidence, $\beta = -0.10$, $SE = 0.03$, $z(25711) = -3.73$, $p < .001$ (see Figures 2b and c). We further investigated this interaction by fitting the following generalized linear mixed model with a logit link function for each confidence level separately: $\text{false alarm (0 or 1)} \sim \text{expected reward} + (1 \parallel \text{participant})$. For the lowest confidence level ("guess"), there was no significant effect of expected reward on false alarms, $\beta = -0.05$, $SE = 0.03$, $z(9356) = -1.70$, $p = .089$. Likewise, there was no significant effect of expected reward on false alarms for the medium confidence level ("sure"), $\beta = -0.01$, $SE = 0.03$, $z(9328) = -0.31$, $p = .753$. However, for the highest confidence level ("very sure"), false alarms were reduced as the reward expectations increased, $\beta = -0.15$, $SE = 0.04$, $t(7027) = -3.27$, $p = .001$. We furthermore investigated the distribution of the number of trials per expected reward level. For false alarms, participants did not expect each reward level equally often, $\chi^2(3) = 1002.01$, $p < .001$ (see Supplementary Figure 1)."

We furthermore report the two alternative versions of the d' analysis we mention in the interim discussion of experiment 1 in the supplement.

2. The authors report effects of duration and confidence in Experiment 1 but don't report effects of duration in Exp 2 or 3 or confidence in Experiment 3. It would be good to report these results for completeness (if only in the text). The authors should also consider adding stimulus duration to their mixed models that evaluate reward since it will account for some variance and this variable could interact with reward. If there is some other reason stimulus duration was manipulated it should be stated; it's unclear from the paper why this was done.

We state in the Method section (and each preregistration) that exposure duration was added to control for effects of exposure strength. This was not an analysis of interest – which is why we also preregistered that all measures will be analysed collapsed across durations – and only done as a sanity check of the paradigm. Feld et al. (2014) previously found that the hit rate increases for longer durations, arguing that encoding strength increases the hit rate. They also find no interaction between reward and duration, which is why they further argue that reward and duration are independent manipulators of hit rate. For future studies involving sleep manipulations, we needed this variable to anticipate whether effects of encoding strength (that should interact with sleep) are present in our modified versions of the paradigm.

For completeness, we now report the duration findings for each experiment. We furthermore have added the following sentence to the Method section:

“After a second fixation (1000-1500 ms), a landscape image was presented, for one of four possible exposure durations (see description of each experiment) to control for encoding strength effects unrelated to reward. This was done as a positive control of our paradigm as well as to prepare future studies that will intervene during the retention period, because exposure duration has previously been found to increase hit rate, without interacting with reward (Feld et al., 2014).”

Since the entire Method section has been moved in front of the results, readers will now be informed about the exposure duration levels before reading the results.

We have added a paragraph about the exposure duration to the results of experiment 2:

“To investigate whether exposure duration influences hit rate, we ran the following generalized linear mixed model with a logit link function for the congruent condition: $\text{hit} \ (\emptyset \text{ or } 1) \sim \text{duration} + (\text{duration} \mid \text{participant}) + (1 \parallel \text{image})$. The longer an image was presented during the learning phase, the more likely it became that participants in the congruent group made a hit in a given target trial, $\beta = 0.14$, $SE = 0.05$, $z(13184) = 2.50$, $p = .012$. Consequently, in the congruent condition, the hit rate for the longest exposure duration of 2500 ms ($M = 0.62$, $SD = 0.11$) was significantly

higher than for the shortest exposure duration of 1500 ms ($M = 0.60$, $SD = 0.12$), $t(102) = 2.36$, $p = .020$, $d_z = 0.23$, 95% $CI [0.04, 0.43]$.

For the incongruent group, we fit a generalized linear mixed model with a logit link function of the same structure as for the congruent condition. The longer an image was presented during the learning phase, the more likely it became that participants in the incongruent group made a hit in a given target trial, $\beta = 0.15$, $SE = 0.06$, $z(13056) = 2.56$, $p = .010$. Consequently, in the incongruent condition, the hit rate for the longest exposure duration of 2500 ms ($M = 0.60$, $SD = 0.16$) was significantly higher than for the shortest exposure duration of 1500 ms ($M = 0.57$, $SD = 0.14$), $t(101) = 2.60$, $p = .011$, $d_z = 0.26$, 95% $CI [0.06, 0.45]$."

We have added a paragraph about exposure duration to the result section of experiment 3:

"The hit rate for the long duration level ($M = 0.63$, $SD = 0.13$) did not significantly differ from the hit rate for the short duration level ($M = 0.64$, $SD = 0.14$), $t(186) = -1.49$, $p = .137$, $d_z = -0.11$, 95% $CI [-0.25, 0.03]$."

We added a model including confidence to the section "Hits" in the results of experiment 1:

"We explored the effect of reward and confidence and their potential interaction in the following generalized linear mixed model with a logit link function: $\text{hit} (0 \text{ or } 1) \sim \text{reward} * \text{confidence} + (\text{reward} + \text{confidence} | \text{participant}) + (\text{confidence} | \text{image})$. There was no significant effect of reward on hits, $\beta = 0.05$, $SE = 0.03$, $z(25600) = 1.67$, $p = .096$, but hits increased as confidence increased $\beta = 0.45$, $SE = 0.07$, $z(25600) = 6.36$, $p < .001$. Additionally, there was a significant interaction of reward and confidence, $\beta = 0.06$, $SE = 0.02$, $z(25600) = 2.70$, $p = .007$. To follow up on the interaction, we ran the following following generalized linear mixed model with a logit link function: $\text{hit} (0 \text{ or } 1) \sim \text{reward} + (1 | \text{participant}) + (1 | \text{image})$, separately for each confidence level. Reward had no significant effect on hits for the lowest confidence level ("guess"), $\beta = 0.05$, $SE = 0.03$, $z(8827) = 1.77$, $p = .076$. However, the higher the reward, the more hits participants achieved when they were "sure", $\beta = 0.09$, $SE = 0.03$, $z(8657) = 2.85$, $p = .004$, or "very sure", $\beta = 0.19$, $SE = 0.04$, $z(8116) = 4.86$, $p < .001$."

We have expanded the analysis of confidence for experiment 2:

"We explored whether confidence affected hits, potentially interacting with reward, in the following generalized linear mixed model with a logit link function for the congruent group: $\text{hit} (0 \text{ or } 1) \sim \text{reward} * \text{confidence} + (\text{reward} + \text{confidence} | \text{participant}) + (\text{confidence} | \text{image})$. The higher their confidence, the more hits participants made, $\beta = 0.55$, $SE = 0.10$, $z(13184) = 5.66$, $p < .001$. However, there was no significant effect of reward on hits, $\beta = 0.08$, $SE = 0.04$, $z(13184) = 1.69$, $p = .090$ (although the p-value suggests a trend in the same direction as without

adding confidence), and no interaction between reward and confidence, $\beta = 0.05$, $SE = 0.04$, $z(13184) = 1.32$, $p = .187$. For the incongruent group, we explored the influence of true reward and confidence on hits in the following generalized linear mixed model with a logit link function: $\text{hit} (0 \text{ or } 1) \sim \text{true reward} * \text{confidence} + (\text{confidence} \mid \mid \text{participant}) + (1 \mid \text{image})$. With increasing confidence, hits increased, $\beta = 0.58$, $SE = 0.09$, $z(13056) = 6.53$, $p < .001$, but there was no significant effect of true reward on hits, $\beta = 0.05$, $SE = 0.04$, $z(13056) = 1.22$, $p = .221$, nor a significant interaction between true reward and confidence, $\beta = -0.02$, $SE = 0.03$, $z(13056) = -0.57$, $p = .572$. We further investigated the influence of shown reward and confidence on hits in the incongruent group with the following generalized linear mixed model with a logit link function: $\text{hit} (0 \text{ or } 1) \sim \text{shown reward} * \text{confidence} + (\text{shown reward} + \text{confidence} \mid \text{participant}) + (\text{confidence} \mid \mid \text{image})$. The higher the shown reward, the more hits participants made, $\beta = 0.19$, $SE = 0.05$, $z(13056) = 3.69$, $p < .001$, and the higher their confidence, the more hits participants made, $\beta = 0.61$, $SE = 0.10$, $z(13056) = 5.99$, $p < .001$. However, there was no significant interaction between shown reward and confidence, $\beta = -0.05$, $SE = 0.04$, $z(13056) = -1.37$, $p = .170$.

We explored whether confidence and reward affected false alarms in the congruent group using the following generalized linear mixed model with a logit link function: $\text{false alarm} (0 \text{ or } 1) \sim \text{shown reward} * \text{confidence} + (\text{shown reward} \mid \text{participant}) + (1 \mid \text{image})$. With increasing confidence, false alarms decreased, $\beta = -0.31$, $SE = 0.05$, $z(13184) = -5.75$, $p < .001$, but there was no significant effect of shown reward on false alarms, $\beta = 0.05$, $SE = 0.05$, $z(13184) = 1.04$, $p = .300$, nor a significant interaction between shown reward and confidence, $\beta = 0.03$, $SE = 0.04$, $z(13184) = 0.71$, $p = .480$. For the incongruent group, we investigated whether confidence and reward affected false alarms using the following generalized linear mixed model with a logit link function: $\text{false alarm} (0 \text{ or } 1) \sim \text{shown reward} * \text{confidence} + (\text{shown reward} + \text{confidence} \mid \text{participant}) + (1 \mid \text{image})$. The higher the shown reward, the more false alarms participants made, $\beta = 0.20$, $SE = 0.06$, $z(13056) = 3.51$, $p < .001$, and the higher their confidence, the less false alarms participants made, $\beta = -0.53$, $SE = 0.10$, $z(13056) = -5.52$, $p < .001$. However, there was no significant interaction between shown reward and confidence, $\beta = 0.01$, $SE = 0.04$, $z(13056) = 0.26$, $p = .794$.”

We added a paragraph about confidence to the result section of experiment 3:

“In an exploratory analysis, we evaluated the influence of confidence and reward on the hit rate and the false alarm rate. We ran the following generalized linear mixed model with a logit link function on the target trials: $\text{hit} (0 \text{ or } 1) \sim \text{confidence} * \text{reward} + (\text{confidence} + \text{reward} \mid \text{participant}) + (\text{confidence} \mid \mid \text{image})$. In this model, we confirmed that the higher the reward, the more hits

participants achieved, $\beta = 0.11$, $SE = 0.05$, $z(23936) = 2.43$, $p = .015$. Furthermore, participants had more hits with increasing confidence, $\beta = 0.92$, $SE = 0.07$, $z(23936) = 13.12$, $p < .001$. There was no significant interaction between reward and confidence, $\beta = -0.01$, $SE = 0.04$, $z(23936) = -0.24$, $p = .810$.

For lure trials, we ran the following generalized linear mixed model with a logit link function: $\text{false alarm (0 or 1)} \sim \text{confidence} * \text{reward} + (\text{confidence} + \text{reward} \mid \mid \text{participant}) + (\text{confidence} \mid \mid \text{image})$. For high rewards, participants made more false alarms, $\beta = 0.10$, $SE = 0.05$, $z(23936) = 2.25$, $p = .024$. Likewise, participants made more false alarms with increasing confidence, $\beta = -0.39$, $SE = 0.08$, $z(23936) = -4.84$, $p < .001$. There was no significant interaction between reward and confidence, $\beta = -0.03$, $SE = 0.05$, $z(23936) = -0.63$, $p = .527$."

Furthermore, we have added duration as a predictor to the key models of each experiment. Since we preregistered that the data would be analysed collapsed across exposure durations, we still report the models without the duration predictor in the main text, and have added the additional models to the Supplementary Materials, along with a discussion of the results.

3. In experiment 2, the effect of reward on memory strength (d') for congruent trials looks very nonlinear across reward values (no change across the first three reward levels and a big change in the final reward level). The authors report a significant linear effect of reward value on d' , but this clearly doesn't describe the effect. Is this effect still significant if the authors allow reward to vary nonlinearly or code it categorically? Given that this is the only significant effect of reward on memory strength reported in the paper, it would be good to ensure robustness. It may also be worth discussing what's driving the effect – there's a *reduction* in false alarm rate from the second highest to highest reward level but an increase in the hit rate. How do the authors interpret this?

We did not run a generalised linear mixed model for d' , because that would require to use a link function describing how the effect should vary (if not linearly). We do not believe that the standard functions that are available (such as binomial or poisson) describe the data well. However, we now report an additional linear mixed model that treats reward as a categorical predictor, which confirms that the effect is driven by the highest reward category:

"Upon visual inspection (see Figure 6a), it appeared that the effect of reward on d' seems to be mainly driven by the highest reward level, while there seemed to be no difference between the lower levels. We followed this up with an additional exploratory linear mixed model of the form $d' \sim \text{reward} + (1 \mid \text{participant})$, where reward was not treated as a linear, but a categorical predictor (because more parameters must be estimated for a categorical predictor, the model structure had to be simplified in order to reach convergence). The lowest reward level was used as reference category.

Indeed, only the highest reward level differed significantly from the first level $\beta = 0.13$, $SE = 0.05$, $t(306.00) = 2.90$, $p = .004$, while the two medium reward categories did not (all $p > .669$)."

We have added a statistical test for the reduction in false alarm rate from the second highest to the highest reward level, which was not significant:

"Descriptively, the false alarm rate drops between the 3rd (1450 gems; $M = 0.37$, $SD = 0.12$) and highest (2150 gems; $M = 0.34$, $SD = 0.12$) reward level, but exploring this difference, we found that it was not statistically significant, $t(203.45) = -1.40$, $p = .163$, $d = -0.19$, 95% $CI [-0.47, 0.08]$."

Nonetheless, we agree that the reduction in the false alarm rate for the highest reward level behind the increased memory strength for the highest reward level, which we have elaborated on in the discussion:

"In our experiment 2, the influence of reward on false alarms differed between the congruent and incongruent group. In the congruent group, higher rewards increased the false alarm rate to a lesser degree than in the incongruent group – in fact, there was a decrease in false alarms between the third (1450 gems) and the fourth (2150 gems) reward level. Given that the reward-related increase in memory strength for the congruent group only emerged for the highest reward level, it is plausible that it is the reduction in false alarms for the highest reward level that is contributing to the effect. That is, both the hit rate and the false alarm rate increase as a function of reward for lower reward levels, reflected in a more lenient decision criterion. However, the reduction in false alarms for the highest reward level results in an increased memory strength for the highest reward level. Since the reward for lures is only presented during the test phase, it is not based on memory, however, the decrease in false alarms for the highest reward category might still be a result of memory processes. Participants in the congruent group could rely on their source memory especially for high reward targets. When encountering a new picture with a high reward, this might have resulted in a greater conflict with their source memory, which made them more careful. Participants in the incongruent group may have been surprised by the incongruence of their memory for stimulus-reward contingencies, and unable to use such strategies. It is relevant to note that such subtle effects cannot be found in experiment 3 that only used two reward levels, which may well explain why we did not find an effect of reward on sensitivity in that experiment. Intriguingly, experiment 3 was also the only one that did not show a clear effect of duration on hit rate, which may point towards the counterintuitive interpretation that using more categories enhances sensitivity to changes in memory processing."

It is worth noting that when we preregistered experiment 1, it was our original intention to test whether the effects of reward were linear or not. We report these analyses in the supplement.

4. I thought that participants with chance level or below chance level d' were excluded, but there are some negative d' values in Figure 5A and 5B. Can the authors clarify?

We have added the following clarification to the figure (now Figure 6):

“Note that participants with an overall memory performance of $d' \leq 0$ were excluded from data analysis. However, participants may still show a performance of $d' \leq 0$ when d' is calculated for each reward level.”

5. I'm confused by the conclusion reached in the abstract and discussion that this paper validates the use of hit rate to study this reward-related phenomenon, e.g. “In this framework, the hit rate, that includes information about decision-making processes, better reflects the processes of interest than pure measures of memory strength such as the sensitivity index”. The authors clearly identify the problem with using hit rate; it reflects both memory strength and decision-making processes and they can't be dissociated. Future studies relying on hit rate will also not be able to interpret results in favor of one or the other. It seems to me that criterion best isolates the process underlying the behavioral effect of interest.

We have elaborated on why we think the hit rate might be preferred as outcome measure in the discussion:

“That is, participants must learn stimulus-reward associations, which are then used to make a decision at test. Along this line, it was recently shown that even memory for unchosen rewards can bias decision making (Biderman & Shohamy, 2021). Thus, pure measures of memory strength (i.e., d') do not isolate the processes underlying the memory-related modulation of performance by reward in the Motivated Learning Task. Instead, measures of the decision criterion and ideally, reward association memory, are preferred. However, modifying the Motivated Learning Task in such a way that these measures can be assessed is not trivial, and alternative versions of the task proposed so far all come with drawbacks. This is why we conclude that the standard version of the task, with the hit rate as outcome measure, might be a valid option in many scenarios. However, it should not be interpreted as a measure of memory strength, but rather as a measure of a memory-based criterion.”

Furthermore, we attenuated the statement in the final conclusion:

“In conclusion, the highly robust effect of reward on hit rate reflects decision criteria at retrieval rather than changes in memory strength during encoding or consolidation. We propose that the decision

criterion at recognition is based on memory for stimulus-reward associations learned during encoding and consolidated thereafter. However, it will be important to develop tasks that can reliably measure memory for stimulus-reward associations and reward effects on memory strength within one task in the future.”

We have also removed the last sentence from the abstract, because it might be misleading without the context of the discussion.

6. In the final paragraph of the discussion the authors state, “We propose that hit rate is a valid measure of associative memory for stimulus-reward pairs learned during encoding and consolidated thereafter.” It is interesting and compelling that the authors show that the effect of reward on hit rate is related to participants’ ability to correctly recall/predict the reward associated with each stimulus (Experiment 1). However, the effect size of this correlation is modest and it seems strange to claim that item recognition memory is a valid measure of associative memory, when we can test associative memory explicitly instead. The authors should consider rephrasing to avoid saying this.

We have rephrased the final conclusion (see above).

7. The authors could be more clear about how their paper goes beyond the Bowen 2020 paper, which reaches a similar conclusion.

We have added the following paragraph to the discussion:

“Our experiments reach similar conclusions as the experiments by Bowen et al. (2020). However, our study goes beyond them since in contrast to their work we are able to examine the effect of associative reward memory and the item specific effects of showing rewards at retrieval, which the authors draw implicit conclusions about without directly investigating them. Detailing on this, we test participants’ source memory for the rewards in experiment 1, which provided crucial insights into participants’ reward expectations. Like Bowen et al. (2020), our version of the task in experiments 2 and 3 provided participant with perfect reward information, minimizing the need for source memory during recognition. What makes our version of the task different is that we were able to manipulate the congruency of the reward information presented during the test phase. We believe that our results complement and go beyond the results of Bowen et al. (2020).” In addition, our large sample allows for a more precise and more robust estimate of effect sizes. We demonstrate that reward effects on memory can be categorized as small (Cohen, 1988), which means that common sample sizes around 30 participants are not enough to detect them reliably and therefore the work by Bowen et al. (2020) was likely not powerful enough to find subtle effects on sensitivity and thus their conclusions based on absence of evidence were premature.”

Minor Comments:

1. The paper would be more clear if the task figure came first, rather than last.

The task figure is now figure 1. Furthermore, the entire Method section has been moved in front of the results, providing more context for the findings.

2. I'm not familiar with the notation "reward || participant" in the lmer models. Is this distinct from "reward | participant"? If so, the authors should clarify and justify when one is used versus the other.

According to the standard notation (see e.g. Singman & Kellen (2019)), "|" vs. "||" denote the following:

- "reward|participant" means that a random slope per participant is estimated for the reward effect, and additionally, a random intercept per participant. Furthermore, the correlation between the two is estimated.
- "reward || participant" is the same as "reward|participant", but minus the correlation. I.e., only a random slope per participant for the reward effect and a random intercept are estimated.

Since adding the correlation means that more parameters need to be estimated for the model, convergence becomes less likely. In the Methods section (under the heading "Statistical analysis"), we describe our model fitting procedure (i.e., when | and || is used), but now have added a sentence referencing this notation style:

"In our model equations, (...|participant) and (...|image) denote | random effects by participant and individual image, respectively (for details on model notation, see Singman & Kellen (2019)). For each analysis, we first tried to fit a maximal model, as recommended by Barr et al. (2013). When a model did not converge or resulted in a singular fit, we reduced it by first removing correlations between random slopes and intercepts. If convergence was still not achieved, we next removed random slopes."

Reviewer 2

As noted in the introduction, one of the main goals of the study is to create a paradigm where memory sensitivity and decision making can be separated without presenting "perfect information" about the reward information at retrieval. While no reward information was presented at test in Exp. 1, these metrics could not be calculated. In the other two experiments, these metrics were calculated, but reward information was also presented at test. It seems one of the main aims of the study to really test these research questions about mechanism was not really achieved. Please provide more discussion of this in the paper.

We agree with the reviewer up to a certain degree. We set out to demonstrate that a bias free measure of reward enhanced memory strength can be achieved, but our studies indicate that this effect, if it exists, is very small. At the same time, our investigation clearly revealed that there is a substantial impact of associative memory for rewards that contributes to participants' reward related behaviour at retrieval. This is the crucial insight that can be gleaned from our research and that is initially somewhat counter-intuitive. We therefore now elaborate on this issue in the discussion:

“Bowen et al. (2020) associated categories with different amounts of reward to give lures an objective value. Their approach provides perfect information of reward contingencies and removes the need to form stimulus-reward associations, since participants rely on perfectly learned category-reward associations. This decouples decision-making during retrieval from memory processes. We tried to design a paradigm without perfect reward information that would still yield a false alarm rate per reward level in experiment 1, but were not successful. In experiment 2 and 3, we presented rewards during the test phase, providing an objective reward-modulated false alarm rate. As the paradigm used by Bowen et al. (2020), this means that participants had (or believed to have) perfect information about reward contingencies, so could base decision on cues presented during retrieval rather than memory for stimulus-reward associations. However, we directly manipulated the reward information presented to the participants, which was either accurate (congruent group) or not (incongruent group). The differences between the congruent and incongruent group provide empirical evidence that reward information presented during retrieval influences participants' behavior. Crucially, our set of studies demonstrates that the reward associations that bias decision making during retrieval are part of the construct that is of interest to researchers studying long-term memory for rewards rather than an unfortunate side effect of the measurement.”

Furthermore, we have added the following paragraph to the newly added limitations section:

“While experiment 1 enabled us to investigate participants' reward expectations and experiments 2 and 3 allowed us to calculate a reward-dependent false alarm rate, we were not able to create a paradigm that solved both problems at the same time. Future research would greatly benefit from tasks that do not need to provide their participants with perfect information about rewards, but still allow researchers to calculate a reward-dependent false alarm rate. A step in this direction might be an elegant two-step paradigm by Jang et al. (2019), who associated reward probabilities with image categories, which changed over the course of the experiment. This provided them with a memory-based false alarm rate per reward probability. They found that reward prediction errors based on reward probability were associated with memory strength, but did unfortunately not report a measure of criterion.”

Also note that following another reviewer's comment, we have added additional analyses of the effect of (expected) reward on false alarms for experiment 1 in the main article. We furthermore report an analysis of memory strength and criterion for experiment 1 in the supplement, which needs to be interpreted with caution due to the limitations of the false alarm rate calculated based on the expected rewards, but might still provide more insight about the results of experiment 1.

Relatedly, one main take-away -- hit rates contain essential information about learning, and how d' doesn't isolate memory processes-- needs more clarification (lines 453-458). It isn't clear to me from the results why this is the conclusion from study 1 given that the false alarm rate information was not usable.

We have elaborated on why we think the hit rate might be preferred as outcome measure in the discussion:

“That is, participants must learn stimulus-reward associations, which are then used to make a decision at test. Along this line, it was recently shown that even memory for unchosen rewards can bias decision making (Biderman & Shohamy, 2021). Thus, pure measures of memory strength (i.e., d') do not isolate the processes underlying the memory-related modulation of performance by reward in the Motivated Learning Task. Instead, measures of the decision criterion and ideally, reward association memory, are preferred. However, modifying the Motivated Learning Task in such a way that these measures can be assessed is not trivial, and alternative versions of the task proposed so far all come with drawbacks. This is why we conclude that the standard version of the task, with the hit rate as outcome measure, might be a valid option in many scenarios. However, it should not be interpreted as a measure of memory strength, but rather as a measure of a memory-based criterion.”

Furthermore, we attenuated the statement in the final conclusion:

“In conclusion, the highly robust effect of reward on hit rate reflects decision criteria at retrieval rather than changes in memory strength during encoding or consolidation. We propose that the decision criterion at recognition is based on memory for stimulus-reward associations learned during encoding and consolidated thereafter. However, it will be important to develop tasks that can reliably measure memory for stimulus-reward associations and reward effects on memory strength within one task in the future.”

We have also removed the last sentence from the abstract, because it might be misleading without the context of the discussion.

Further, Experiment 3 is most similar to the studies that the authors are comparing to, and the findings replicate prior work with no effect on memory sensitivity but effects on response bias, but this is glossed over in the general discussion.

We have elaborated on this in the discussion:

“Our third experiment with only two reward categories and perfect information about the rewards at recognition is most comparable to prior work that – as we did in experiment 3 – find that reward does not influence memory strength, but only decision criterion. Two interpretations are possible: There might be an effect of reward on memory strength, but it could be so small that it cannot be detected reliably with even large sample sizes as in our study (around $N = 100$), let alone with “standard” sample sizes in the field (around $N = 30$). This would make it very hard to assess these effects in more resource-heavy neuroscientific experiments. The alternative explanation is that effects of reward on memory strength only arise under very specific conditions. For example, the reward-based change in memory strength in our second experiment was largely driven by the highest reward level, while the lower reward levels did not differ in the effect on memory strength. It might be the case that effects of reward on memory strength only emerge when there are at least more than two reward levels, allowing participants to focus on the comparably few trials with a very high reward, as compared to half of the trials with a high reward in the case of two reward levels. This potential moderator of reward context should be tested in future research.”

In reference to the effect of reward on memory strength being driven by the highest reward category, also see the updated results of experiment 2 and the discussion of the results (response to reviewer 1).

It seems to me that the main conclusion from the results is that when no information about reward is explicitly presented at retrieval participants may rely more on memory, but when information is presented at retrieval, this overrides memory strength and participants rely on decision biases. This is touched on by the authors but is sort of buried with the discussion.

We are unsure what the reviewer refers to. In experiment 2, participants in the congruent and incongruent group still rely on memory strength: Since the sensitivity measure d' allows to differentiate between memory strength and bias-related behaviour of the participants, the overall positive d' in experiment 2 indicated that participants were still using the memory traces they formed during learning.

However, the reviewer is correct to point out that in the incongruent group, the mismatch between the true reward and the shown reward has the potential to disrupt memory strength. To investigate this possibility, we added exploratory analyses to the result section of

experiment 2. In these analyses, we wanted to find out whether participants' memory performance is associated with the degree to which they are influenced by the shown reward in the incongruent group. We find that participants who are more strongly influenced by the shown reward information show worse memory strength, indicating that those who are able to ignore the conflicting reward information perform better (also see the newly added Figure panels 4d and 4e):

“In an exploratory analysis, we investigated whether the influence of the shown reward on participants' decisions was associated with their overall memory performance. First, we extracted the by-participant slope for the shown reward effect from a mixed model of the form $\text{response (old or new)} \sim \text{shown reward} + (\text{shown reward} \mid \text{participant}) + (\text{shown reward} \mid \text{image})$. We then correlated the (absolute) slope with participants' overall memory strength in the task (d'). Participants whose old/new decisions were more strongly influenced by the shown reward (i.e., the more the slope differed from 0) performed worse in the Motivated Learning Task, $r(100) = -.31, p = .002$ (see Figure 4d). Upon visual inspection, it seemed like only a few data points belonging to participants with steeper slopes were driving the correlation. However, when removing those outliers (all participants with a slope $\geq .3$), the correlation remained significant, $r(89) = -.23, p = .030$. The shown reward might influence target trials differently, because for targets, shown reward information might be in conflict with the true reward information. This is why we repeated the previous exploratory analysis for target trials. We extracted the by-participant slope for the shown reward effect from a mixed model of the form $\text{hit (0 or 1)} \sim \text{shown reward} + (\text{shown reward} \mid \text{participant}) + (1 \mid \text{image})$. We then correlated the (absolute) slope with participants' overall memory strength in the task (d'). Participants whose hits were more strongly influenced by the shown reward (i.e., the more the slope differed from 0) performed worse in the Motivated Learning Task, $r(100) = -.29, p = .003$ (see Figure 4e). Upon visual inspection, it seemed like only a few data points belonging to participants with steeper slopes were driving the correlation. However, when removing those outliers (all participants with a slope $\geq .5$), the correlation remained significant, $r(95) = -.20, p = .047$.”

We elaborate on this intriguing thought in more detail in the discussion:

“It is possible that providing participants with reward information during recognition changes the cognitive processes participants use to complete the task. I.e., when participants are not presented with any reward information during recognition (experiment 1), they might rely more strongly on (stimulus-reward) memory. When reward information is provided during recognition (experiments 2 and 3), participants might shift their criterion based on the presented reward, and disregard previously

formed (stimulus-reward) memory. While we cannot rule out this explanation entirely, we argue against it: Overall memory performance is descriptively even better in experiments 2 and 3 than in experiment 1 (see Supplementary Table 4), and better performance can only be achieved by improved discrimination between old and new pictures. Furthermore, reward information presented during recognition is not neutral, but seems to influence memory performance: For participants in the incongruent group, memory strength is worse when they are more strongly influenced by the shown reward. In line with this, response patterns in experiment 2 differ between the congruent and the incongruent group, especially for false alarms. This suggests that even when reward information is presented during recognition, knowledge about stimulus-reward associations is used to retrieve matching memory traces. If the reward information presented during recognition would override the participants' memory entirely, there would be no difference in performance between the congruent and incongruent group. We argue that rather than overriding existing memory traces, the reward information presented during recognition is used as a memory cue that guides retrieval. Lastly, it would be maladaptive if participants disregarded their memory of the pictures in favor of the reward information presented during recognition."

The rationale for different exposure durations during encoding, and why analysis of this only occurred in Experiment 1, is also not clear.

In the method section (and our preregistrations), we explain that duration was added to control for effects of exposure strength. Because this was not an analysis of interest, we furthermore preregistered that all our main analyses will be collapsed across durations. Feld et al. (2014) report that reward and exposure duration do not interact, and argue that reward and duration affect hit rate independently. We included duration as a variable because we plan future studies investigating sleep effects, that should interact with encoding strength. Thus, we needed to ensure that our modified version of the paradigm is still sensitive to a manipulation of exposure duration.

For completeness, we now report the duration findings for each experiment. We furthermore have added the following sentence to the Method section:

"After a second fixation (1000-1500 ms), a landscape image was presented, for one of four possible exposure durations (see description of each experiment) to control for encoding strength effects unrelated to reward. This was done as a positive control of our paradigm as well as to prepare future studies that will intervene during the retention period, because exposure duration has previously been found to increase hit rate, without interacting with reward (Feld et al., 2014)."

Since the entire Method section has been moved in front of the results, readers will now be informed about the exposure duration levels before reading the results.

We have added a paragraph about the exposure duration to the results of experiment 2:

“To investigate whether exposure duration influences hit rate, we ran the following generalized linear mixed model with a logit link function for the congruent condition: $\text{hit} \ (0 \text{ or } 1) \sim \text{duration} + (\text{duration} \mid \text{participant}) + (1 \mid \mid \text{image})$. The longer an image was presented during the learning phase, the more likely it became that participants in the congruent group made a hit in a given target trial, $\beta = 0.14$, $SE = 0.05$, $z(13184) = 2.50$, $p = .012$. Consequently, in the congruent condition, the hit rate for the longest exposure duration of 2500 ms ($M = 0.62$, $SD = 0.11$) was significantly higher than for the shortest exposure duration of 1500 ms ($M = 0.60$, $SD = 0.12$), $t(102) = 2.36$, $p = .020$, $d_z = 0.23$, 95% CI [0.04, 0.43].

For the incongruent group, we fit a generalized linear mixed model with a logit link function of the same structure as for the congruent condition. The longer an image was presented during the learning phase, the more likely it became that participants in the incongruent group made a hit in a given target trial, $\beta = 0.15$, $SE = 0.06$, $z(13056) = 2.56$, $p = .010$. Consequently, in the incongruent condition, the hit rate for the longest exposure duration of 2500 ms ($M = 0.60$, $SD = 0.16$) was significantly higher than for the shortest exposure duration of 1500 ms ($M = 0.57$, $SD = 0.14$), $t(101) = 2.60$, $p = .011$, $d_z = 0.26$, 95% CI [0.06, 0.45].”

We have added a paragraph about exposure duration to the result section of experiment 3:

“The hit rate for the long duration level ($M = 0.63$, $SD = 0.13$) did not significantly differ from the hit rate for the short duration level ($M = 0.64$, $SD = 0.14$), $t(186) = -1.49$, $p = .137$, $d_z = -0.11$, 95% CI [-0.25, 0.03].”

Reviewer 3

Further comments:

1. As just stated, I was surprised that RT data were not analyzed. If I understand the authors correctly, the idea of the “strategy” hypothesis is that, if participants recall the associated (larger) reward of an old item at retrieval, they “try harder” to be correct to get the larger reward. This strategy adaptation would in my view imply that participants take more time to make their decision, in the sense of an increased threshold / boundary separation in a sequential sampling model / drift-diffusion model (DDM). It would be interesting to see whether there is evidence for this effect. Because reward could have additional effects (including a change in sensitivity), the best approach would obviously be to apply these models to the data. In my view, a change in sensitivity (similar to d' in SDT) would be related to a change in drift rate, whereas a change in strategy (similar to criterion in SDT) would be related to a change in the threshold and/or starting point. The advantage of SSM/DDM over SDT is, that the RT data might contain very critical information to answer this question.

We conducted complementary drift diffusion models for all three experiments. Since these are additional analyses that were not preregistered, we refrain from interpreting the results

in the main manuscript, but instead report and discuss the analyses in the Supplementary Materials (section: “Alternative Analysis: Drift Diffusion Models”) for interested readers.

2. In the sub-section “Expected Reward” of the results of the first experiment, an analysis of confidence rating is reported, but it has never been mentioned before that confidence ratings were obtained. This should be mentioned earlier (when explaining the task design), and the rationale for including (and analyzing) them should be given.

We agree that mentioning confidence only in the methods, but not before the results, is confusing. We have added the following sentence to the description of experiment 1:

“[...] Participants learned several pictures associated with different amounts of points (gems), which were earned for correctly identifying the pictures in a recognition memory test 24 later. After identifying a picture as either old or new, we asked participants for their confidence in their decision (“guess”, “sure”, “very sure”). Previous studies using the Motivated Learning Task have found that high confidence responses are sensitive to reward, but found no evidence for (or against) a reward effect in low confidence items (Adcock et al., 2006). [...]”

We now also mention confidence explicitly in the descriptions of experiment 2 and 3. As requested by reviewer 1, the figure including the task design has been moved to the beginning of the manuscript, which should provide additional clarification about where the confidence ratings are included.

Relatedly, in the same section the false alarm result of Figure 1B is reported, but this should be reported earlier on (the authors jump back and forth

In response to a comment by reviewer 1 and this comment, the result section for experiment 1 has been restructured. We now report additional analyses for the false alarms, which have been moved to a separate section about false alarms. As a result, the former Figure 1B has been moved to the Supplementary Materials, and has been replaced by the more informative false alarm figures. (Note that the former Figure 1 is now Figure 2, since the task graphic has been moved to the beginning of the manuscript in response to a reviewer comment.)

3. “For participants whose reward expectations were more accurate, the hit rate was more strongly influenced by true reward, $r(198)=.24$, $p<.001$ (Figure 2C).” Is this still one of the preregistered hypotheses? Relatedly, the analysis for Figure 2D feels highly circular/redundant to the analysis for Figure 2C

As stated in the manuscript, the relationship between the accuracy of reward expectations and the effect of reward on the hit rate were exploratory.

The difference between the analyses for Figures 2C and 2D (now Figures 3C and 3D) is subtle, but they measure related, but different concepts. Analysis C focuses on whether the reward

expectations are correct or not, based on a binary classification. That is, it captures whether participants can accurately remember the reward information, but ignores it when participants are “close” (e.g., for a true reward of 2150, reward expectations of 1450 and 50 are equally wrong, even though 1450 is closer to the truth). Analysis D focuses on the degree in which reward expectations scale with the true reward. Crucially, when reward expectations scale as a function of the true reward, that does not necessarily imply that participants’ reward expectations are correct. It is possible that a participant expects higher rewards as the true reward increases, but still does not expect the correct reward value. That is, when a participant correctly identifies the exact reward values, their reward expectations will also scale as a function of the true reward, but not necessarily vice versa. This is especially relevant since participants were biased towards choosing the medium reward categories.

In our case, both analyses lead to highly similar results, but it would have been possible that e.g., only exact reward representations (binary classification as correct or not) drive the effect of reward on hit rate. In this case, the relationship for analysis C would have been stronger than for analysis D.

We have added more details to the description of both analyses in the result section of experiment 1 to highlight the difference between them:

“We next explored whether more accurate reward expectations were related to a larger effect of true reward on the hit rate. To this end, we correlated the per-participant slopes for the reward effect from the `hit ~ reward` model with the % of correct reward expectations for hits per participant. This quantifies the relationship between the proportion of accurate reward expectations (binary classification) and how strongly the hit rate is influenced by the true reward. For participants whose reward expectations were more accurate, the hit rate was more strongly influenced by the true reward, $r(198) = .24, p < .001$ (Figure 3c). Additionally, we calculated the correlation with the per-participant slopes for the reward effect from the `hit ~ reward` model with the per-participant slopes for the reward effect from an `expected_reward ~ reward` model (`expected_reward ~ reward + (reward|participant)` for hits). Other than the previous analysis, this analysis does not only focus on correct reward expectations, but acknowledges that a participant expecting 1450 when the true reward is 2150 is closer to the truth than a participant expecting 50 gems. Participants whose reward expectations were more strongly influenced by the true reward showed a larger influence of reward on the hit rate, $r(198) = .26, p < .001$ (see Figure 3d).”

4. "Since it is unclear what the false alarm rate based on the expected reward represents" Plotting the false alarm rate as a function of expected reward would be informative.

We have added Figure 3B and 3C as a visual representation of the relationship between reward expectations and false alarms. (Also see the newly added section "false alarms" in the result section of experiment 1).

2nd Nov 23

Dear Ms Nagel,

Thank you for your patience during the peer-review process. Your manuscript titled "Memory for Rewards Guides Retrieval" has now been seen by the same 3 reviewers as before, and I include their comments at the end of this message.

The reviewers continue to be supportive of your work, but especially Reviewer #3 highlights some important remaining concerns. We remain interested in the possibility of publishing your study in Communications Psychology, but would like to consider your responses to these concerns and assess a revised manuscript before we make a final decision on publication.

We therefore invite you to revise and resubmit your manuscript, along with a point-by-point response to the reviewers. Please highlight all changes in the manuscript text file.

As you prepare your revision, please follow our editorial guidance on the reporting of analyses and results, for pre-registered and exploratory work.

All preregistered analyses should be in the main manuscript, and no preregistered analysis should be omitted unless it has become evident that the analysis is unfeasible or inappropriate for the data. Your manuscript is presently compliant with this guidance.

We also expect that all critical analyses to be included in the main manuscript. This includes non-preregistered analyses, which must be labeled as exploratory. In other words, analyses identified by the referees as important tests of the soundness of the methodological approach or as required to provide stronger evidence for claims in the manuscript must be included and appropriately labeled. We place no restrictions on the length of the Methods section and are flexible about the length of the Results, as we prioritize comprehensive and transparent reporting over manuscript length. Only analyses that are exploratory in the sense that they do not inform the research question but shed light on tangential matters and those that offer "sanity checks" on the data should be relegated to the SI.

Please also note that your revised manuscript must comply with our formatting and reporting requirements, which are summarized on the following checklist: Communications Psychology formatting checklist and also in our style and formatting guide Communications Psychology formatting guide .

Please use the following link to submit your revised manuscript, point-by-point response to the referees' comments (which should be in a separate document to any cover letter) and the completed checklist:

[link redacted]

** This url links to your confidential home page and associated information about manuscripts you may have submitted or be reviewing for us. If you wish to forward this email to co-authors, please

delete the link to your homepage first **

Please do not hesitate to contact me if you have any questions or would like to discuss these revisions further. We look forward to seeing the revised manuscript and thank you for the opportunity to review your work.

Best regards,

Marike

Marike Schiffer, PhD
Chief Editor
Communications Psychology

EDITORIAL POLICIES AND FORMATTING

Editorial Policy: Policy requirements (Download the link to your computer as a PDF.)

* **CODE AVAILABILITY:** All Communications Psychology manuscripts must include a section titled "Code Availability" at the end of the methods section. In the event of publication, we require that the custom analysis code supporting your conclusions is made available in a publicly accessible repository; at publication, we ask you to choose a repository that provides a DOI for the code; the link to the repository and the DOI will need to be included in the Code Availability statement. Publication as Supplementary Information will not suffice. We ask you to prepare code at this stage, to avoid delays later on in the process.

* DATA AVAILABILITY:

All Communications Psychology manuscripts must include a section titled "Data Availability" at the end of the Methods section or main text (if no Methods). More information on this policy, is available at <http://www.nature.com/authors/policies/data/data-availability-statements-data-citations.pdf>.

At a minimum the Data availability statement must explain how the data can be obtained and whether there are any restrictions on data sharing. Communications Psychology strongly endorses open sharing of data. If you do make your data openly available, please include in the statement:

We recommend submitting the data to discipline-specific, community-recognized repositories, where possible and a list of recommended repositories is provided at <http://www.nature.com/sdata/policies/repositories>.

If a community resource is unavailable, data can be submitted to generalist repositories such as figshare or Dryad Digital Repository. Please provide a unique identifier for the data (for example a DOI or a permanent URL) in the data availability statement, if possible. If the repository does not provide identifiers, we encourage authors to supply the search terms that will return the data. For data that have been obtained from publicly available sources, please provide a URL and the specific data product name in the data availability statement. Data with a DOI should be further cited in the methods reference section.

REVIEWERS' COMMENTS:

Reviewer #1 (Remarks to the Author):

This is a revision of a manuscript from Nagel and colleagues that describes several experiments investigating the role of reward on episodic memory decisions. The revisions are responsive to my previous comments. Previous conceptual and technical points of confusion are now clarified and the authors have added a number of supplementary analyses that extend their findings.

Reviewer #2 (Remarks to the Author):

The authors have done a very thorough job of revising the paper. I think the additional analyses and more detailed explanations strengthened the paper.

A few minor comments

Line 291 the word "following" is repeated

line 389 "the" is repeated.

Reviewer #3 (Remarks to the Author):

The authors have done a very good job addressing my remarks on their original submission. However, I have some remaining points about their implementation of the DDM:

1. In the manuscript, it only says "A reviewer suggested drift diffusion models as an alternative analysis strategy. Since these analyses were not preregistered, we refrain from interpreting them in the main manuscript, and instead report them in the Supplementary Materials (section: "Alternative Analysis: Drift Diffusion Models") for interested readers."

I can understand that the authors may not want to add these analyses to their main text. However, I think that at least the rationale of my suggestion should be stated here: A strategy shift in retrieval would be indicated by participants responding more cautiously in high-reward conditions. This motivates the application of the DDM as RT are taken into account and cautiousness is implemented as a parameter of the model (i.e., the boundary separation). Furthermore, it should be stated that the DDM analyses are - generally speaking - in line with this rationale.

2. The authors say that a BIC/AIC difference of >2 is required to support the more complex model. This does not make sense, as the BIC/AIC already account for complexity. Thus, a difference of >0 is sufficient. If the authors distrust these criteria (some argue that they are too lenient), then they need to select other(s).

3. My biggest concern is about the decision to fix the NDT across reward levels. This is problematic, as it is known that RT go down with increasing rewards/values (called "Magnitude" or "Overall Value" Effect; see for instance work by Angelo Pirrone). It has been argued that this RT effect could be due to response vigour, which would map onto the NDT parameter in the DDM. In particular in the first experiment, the unexpected direction of the boundary separation effect could look different, if NDT is allowed to differ across reward conditions. How do the results look like when the NDT parameter is not fixed?

4. There is a typo on page 56: "That is, boundary separation for the highest reward level ($M=2.21$, $SD=0.36$) was significantly higher than the starting point for ..." -> replace "starting point" with "boundary separation"

5. Page 62: "..., which results in a stable drift rate". Drift rates are always stable (at least in the 4-parameter DDM). I think what you want to say is "...but they take more time to make a decision when the reward is high, which results in more integration of evidence and thus more accurate decisions."

Signed,
Sebastian Gluth

Any changes in the manuscript and the supplementary material have been highlighted in yellow. The relevant sections of the manuscript have also been pasted below the respective reviewer comment for convenience.

We fixed the two typos pointed out by reviewers 2 and 3.

Reviewer 3

1. In the manuscript, it only says "A reviewer suggested drift diffusion models as an alternative analysis strategy. Since these analyses were not preregistered, we refrain from interpreting them in the main manuscript, and instead report them in the Supplementary Materials (section: "Alternative Analysis: Drift Diffusion Models") for interested readers."

I can understand that the authors may not want to add these analyses to their main text. However, I think that at least the rationale of my suggestion should be stated here: A strategy shift in retrieval would be indicated by participants responding more cautiously in high-reward conditions. This motivates the application of the DDM as RT are taken into account and cautiousness is implemented as a parameter of the model (i.e., the boundary separation). Furthermore, it should be stated that the DDM analyses are - generally speaking - in line with this rationale.

We have added the rationale for the DDM analyses to the section "Statistics and reproducibility":

"A reviewer suggested drift diffusion models as an alternative analysis strategy, because drift diffusion models take into account reaction times. This allows for a quantification of "cautiousness" (in form of the parameter "boundary separation"), which could e.g. be expected to increase with reward. Since these analyses were not preregistered, we refrain from interpreting them in the main manuscript. Instead, we give a short summary of the findings in the result section of each experiment, and report the drift diffusion models in detail in the Supplementary Materials (section: "Alternative Analysis: Drift Diffusion Models") for interested readers."

Furthermore, we now provide a summary of the DDM results in the results section of each experiment, and additionally, a brief paragraph pointing towards the DDM results in the discussion.

Results Experiment 1:

“For experiment 1, we investigated whether participants were more likely to be correct or not depending on their reward expectations for trials where they responded “old” (see Supplementary Tables 51 – 54 and Supplementary Figure 5). We found no effect of reward on boundary separation and starting point. However, the drift rate increased as the expected reward increased, and there was a trend towards a decreasing non-decision time as the reward increased.”

Results Experiment 2:

“For experiment 2, we investigated whether participants were more likely to respond “old” or “new” as a function of shown reward (see Supplementary Tables 55 – 62 and Supplementary Figures 6 – 7). For the congruent group, we found no significant effect of reward on boundary separation. The starting point decreased and the non-decision time increased as the reward increased. There was a trend towards a decreasing drift rate as the reward increased. For the incongruent group, we found no significant effect of reward on boundary separation. The starting point and the drift rate decreased as the reward increased. The non-decision time increased as the reward increased.”

Results Experiment 3:

“For experiment 3, we investigated whether participants were more likely to respond “old” or “new” as a function of reward (see Supplementary Figure 8). There was no significant effect of reward on boundary separation, starting point, and non-decision time. However, the drift rate for the high reward level was significantly lower than the drift rate for the low reward level.”

Discussion:

“Following the insightful suggestion of a reviewer, we report the results of drift diffusion models as an alternative analysis in the Supplementary Material. While our conclusions are based on our preregistered mixed model analyses, it is helpful to look at the data from a different perspective. The drift diffusion models, which take reaction times into account, generally confirm our findings that reward influences participants’ performance at retrieval. Briefly, our interpretation of the drift diffusion results is that reward cues shown during a recognition trial determine how much effort participants spend on both non-decision related and decision-related processes, regardless of the memory strength of the respective stimulus. On the other hand, when no reward information is presented, participants do not waste time on stimuli that do not have a strong representation in memory (see Supplementary Material section “Alternative Analysis: Drift Diffusion Models” for an in-depth analysis and discussion).”

2. The authors say that a BIC/AIC difference of >2 is required to support the more complex model. This does not make sense, as the BIC/AIC already account for complexity. Thus, a difference of >0 is sufficient. If the authors distrust these criteria (some argue that they are too lenient), then they need to select other(s).

We now use a difference of > 0 as criterion.

3. My biggest concern is about the decision to fix the NDT across reward levels. This is problematic, as it is known that RT go down with increasing rewards/values (called "Magnitude" or "Overall Value" Effect; see for instance work by Angelo Pirrone). It has been argued that this RT effect could be due to response vigour, which would map onto the NDT parameter in the DDM. In particular in the first experiment, the unexpected direction of the boundary separation effect could look different, if NDT is allowed to differ across reward conditions. How do the results look like when the NDT parameter is not fixed?

Thank you for this valuable feedback – we had no hypotheses regarding the NDT, which is why we initially set it as fixed parameter. However, when adding NDT as free parameter, the results change substantially (e.g., there is no effect of reward on boundary separation anymore), and the model fit of the complex model improves. We have updated the results and discussion of the DDM section in the Supplementary Materials and the summaries pasted above reflect these changes.

Interestingly, our findings are opposite to what would be expected based on the magnitude sensitivity phenomenon. We briefly discuss this in the Supplementary Material (e.g., that in contrast to typical magnitude sensitivity paradigms, our task is memory based), but provide a more detailed analysis (beyond the scope of the Supplementary Material) here:

We thought it would be interesting to analyse the data of the incongruent group from the perspective of a magnitude effect by investigating whether the sum of the true and shown reward influences participants' decisions. However, it appears that performance in the incongruent group is largely driven by the shown reward, not by the sum of both rewards: A mixed model of the form $\text{hit} (0 \text{ or } 1) \sim \text{reward sum} + \text{shown reward} + (1 \mid \text{participant}) + (1 \mid \text{image})$ resulted in a significant effect of shown reward, but no significant effect of reward sum, indicating that the variance in hits is mainly explained by the

shown reward, and that the sum of the shown and true reward does not explain any additional variance. This interpretation is supported by a visual representation of how the reward sum, the true and the shown reward relate to the hit rate:

Each panel of the plot shows a different sum of true and shown reward (e.g. a true reward 50 + shown reward 750 = reward sum 800). The x axis shows the true reward, i.e., the shown reward can be inferred from the reward sum. Across the panels, with increasing reward sum, the hit rate increases. However, within each panel, the hit rate decreases as the true reward increases. That is because when the true reward is high, the shown reward must be low, and since the shown reward drives the effect, the hit rate is lower for true rewards within each reward sum panel.

5. Page 62: "... , which results in a stable drift rate". Drift rates are always stable (at least in the 4-parameter DDM). I think what you want to say is "...but they take more time to make a decision when the reward is high, which results in more integration of evidence and thus more accurate decisions."

Since the results of the DDMs have changed, our interpretation and discussion has also changed, the respective sentence has been deleted.

3rd Jan 24

Dear Ms Nagel,

Thank you for submitting a revised version of your manuscript titled "Memory for Rewards Guides Retrieval". After careful consideration and discussion with my colleagues, I am sorry to have to tell you that we do not feel that the editorial requests and Reviewers' comments have been sufficiently addressed to justify sending this revision back to the reviewers.

We take this unusual course of action occasionally to avoid unproductive rounds of review that ultimately reduce the chances of the manuscript obtaining a fair and objective evaluation.

For us to consider this manuscript further please do your best to fully address all of the previously stated requests.

In particular, we ask that the following prioritized list of concerns be comprehensively addressed:

In the last decision letter, we specified the editorial request that the analyses that address the reviewer's concerns be included in the main manuscript, labeled as exploratory analyses. Instead, the results are currently reported in the SI. The SI is reserved for ancillary analyses. Please include the results in the manuscript as previously requested (including full reporting of statistics and turning the SI display items into main Article display items).

Second, the manuscript must at this stage comply with our formatting requirements, including the presentation and order of sections (Abstract, Introduction, Methods, Results, Discussion). Results reporting experiment-by-experiment is helpful, but please remove the "Interim Discussions".

If you can adequately respond to these concerns, we would be happy to look at a revised manuscript.

We hope to receive your revised version as soon as possible. If you anticipate a delay of more than three months, however, please let us know.

If you are not interested in submitting a suitably revised manuscript in the future please let me know immediately so we can close your file. If you have any questions, please contact me.

Please use the link below to submit a suitably revised manuscript and updated response to referees when they are ready.

[link redacted]

In the meantime, I regret that we remain unable to publish your current manuscript in Communications Psychology. We would naturally understand, therefore, if you decided to submit your manuscript elsewhere.

I am sorry that we cannot be more positive as things stand.

Best regards,

Marike

Marike Schiffer, PhD
Chief Editor
Communications Psychology

Any changes in the manuscript have been highlighted in yellow. The relevant sections of the manuscript have also been pasted below the respective reviewer comment for convenience.

General points

- The interim discussions for each experiment have been merged into the general discussion.
- The information about the analysis software can now be found under the heading “code availability”, instead of the heading “statistics and reproducibility”.

We fixed the two typos pointed out by reviewers 2 and 3.

Reviewer 3

1. In the manuscript, it only says "A reviewer suggested drift diffusion models as an alternative analysis strategy. Since these analyses were not preregistered, we refrain from interpreting them in the main manuscript, and instead report them in the Supplementary Materials (section: “Alternative Analysis: Drift Diffusion Models”) for interested readers."

I can understand that the authors may not want to add these analyses to their main text. However, I think that at least the rationale of my suggestion should be stated here: A strategy shift in retrieval would be indicated by participants responding more cautiously in high-reward conditions. This motivates the application of the DDM as RT are taken into account and cautiousness is implemented as a parameter of the model (i.e., the boundary separation). Furthermore, it should be stated that the DDM analyses are - generally speaking - in line with this rationale.

We have added the rationale for the DDM analyses to the section “Statistics and reproducibility”:

“A reviewer suggested drift diffusion models as an alternative analysis strategy, because drift diffusion models take into account reaction times. This allows for a quantification of “cautiousness” (in form of the parameter “boundary separation”), which could e.g. be expected to increase with reward.”

Furthermore, the DDM results have been moved to the main manuscript.

Results Experiment 1:

“For experiment 1, we investigated whether the expected reward had an effect on whether participants made a hit or a false alarm. That is, we ran a drift diffusion model on all trials where participants responded that a picture was “old” (i.e., where data about reward expectations was available), with the possible outcomes “hit” (upper boundary) or “false alarm” (lower boundary). For the complex model, we let boundary separation, starting point, the drift rate, and the non-decision time vary by expected reward level (50, 750, 1450 or 2150 gems). After excluding participants with less than 10 trials per condition, the model was run on $n = 161$ participants. According to the BIC, the complex model was preferred in 100.00 % of cases. According to the AIC, the complex model was preferred in 100.00 % of cases. According to the likelihood ratio test, the complex model provided a significantly better fit in 49.69 % of cases.

To estimate the effect of reward on boundary separation α , we ran a linear mixed model of the form $\alpha \sim \text{expected reward} + (\text{expected reward} \mid \text{participant})$. There was no effect of reward on boundary separation, $\beta = -0.02$, $SE = 0.03$, $t(160.00) = -0.72$, $p = .472$ (see Figure 4a). That is, there was no difference in boundary separation between the highest expected reward level ($M = 2.04$, $SD = 0.89$) and the lowest expected reward level ($M = 2.06$, $SD = 0.39$), $t(160) = -0.28$, $p = .777$, $d_z = -0.02$, 95% $CI [-0.18, 0.13]$. To estimate the effect of reward on the starting point β , we ran a linear mixed model of the form $\beta \sim \text{expected reward} + (\text{expected reward} \mid \text{participant})$. The expected reward did not significantly affect the starting point, $\beta = 0.00$, $SE = 0.01$, $t(160.00) = -0.65$, $p = .513$ (see Figure 4b). That is, the starting point for the highest expected reward level ($M = 0.52$, $SD = 0.12$) was not significantly different from the starting point for the lowest expected reward level ($M = 0.53$, $SD = 0.11$), $t(160) = -0.75$, $p = .455$, $d_z = -0.06$, 95% $CI [-0.21, 0.10]$. However, a linear mixed model of the form $\delta \sim \text{expected reward} + (\text{expected reward} \mid \text{participant})$ revealed that the drift rate increased as the expected reward increased, $\beta = 0.17$, $SE = 0.02$, $t(160.00) = 8.27$, $p < .001$ (see Figure 4c). That is, the drift rate for the highest expected reward level ($M = 0.45$, $SD = 0.48$) was significantly higher than the drift rate for the lowest expected reward level ($M = 0.11$, $SD = 0.35$), $t(160) = 7.52$, $p < .001$, $d_z = 0.59$, 95% $CI [0.42, 0.76]$. A linear mixed model of the form $\tau \sim \text{expected reward} + (\text{expected reward} \mid \text{participant})$ showed a trend towards a decreasing non-decision time as the reward increased, $\beta = -0.02$, $SE = 0.01$, $t(160.00) = -1.88$, $p = .062$ (see Figure 4d). That is, the non-decision time for the highest expected reward level ($M = 1.04$, $SD = 0.24$) was significantly lower than for the lowest expected reward level ($M = 1.08$, $SD = 0.27$), $t(160) = -2.00$, $p = .048$, $d_z = -0.16$, 95% $CI [-0.31, 0.00]$. In general, participants median reaction time (in seconds) was shorter when they expected a high ($M = 1.76$, $SD = 0.39$) versus low reward ($M = 1.94$, $SD = 0.52$), $t(160) = -7.50$, $p < .001$, $d_z = -0.59$, 95% $CI [-0.76, -0.42]$.”

Results Experiment 2:

“For the congruent group of experiment 2, we investigated whether the reward had an effect on whether participants responded “old” or “new. That is, we ran a drift diffusion model on all trials with the outcome variable “old” (lower boundary) or “new” (upper boundary). For the complex model, we let boundary separation, the starting point, the drift rate, and the non-decision time vary by reward level (50, 750, 1450 or 2150 gems). Since no participants had to be excluded due to a lack of trials, the model was run on all $n = 103$ participants. When using the difference in BIC as criterion, the complex model was preferred in 100.00 % of cases. When using the difference in AIC as criterion, the complex model was preferred in 100.00 % of cases. According to a likelihood ratio test between the two models, the complex model provided a significantly better fit in 45.63 % of cases.

To estimate the effect of reward on boundary separation α , we ran a linear mixed model of the form $\alpha \sim \text{reward} + (\text{reward} \mid \text{participant})$. There was no significant effect of reward on boundary separation, $\beta = 0.00$, $SE = 0.01$, $t(102.00) = 0.45$, $p = .656$ (see Figure 8a). That is, there was no significant difference between boundary separation for the highest reward level ($M = 2.06$, $SD = 0.36$) and the lowest reward level ($M = 2.05$, $SD = 0.35$), $t(102) = 0.51$, $p = .608$, $d_z = 0.05$, 95% CI $[-0.14, 0.24]$. To estimate the effect of reward on the starting point β , we ran a linear mixed model of the form $\beta \sim \text{reward} + (\text{reward} \mid \text{participant})$. The starting point decreased as the reward increased, $\beta = -0.01$, $SE = 0.00$, $t(102.00) = -3.14$, $p = .002$ (see Figure 8b). That is, the starting point for the highest reward level ($M = 0.48$, $SD = 0.08$) was significantly lower (closer to “old”) than the starting point for the lowest reward level ($M = 0.51$, $SD = 0.08$), $t(102) = -3.24$, $p = .002$, $d_z = -0.32$, 95% CI $[-0.52, -0.12]$. A linear mixed model of the form $\delta \sim \text{reward} + (\text{reward} \mid \text{participant})$ revealed a trend towards a decreasing drift rate as the reward increased, $\beta = -0.02$, $SE = 0.01$, $t(102.00) = -1.85$, $p = .068$ (see Figure 8c). However, the drift rate for the highest reward level ($M = 0.06$, $SD = 0.23$) did not differ from the drift rate for the lowest reward level ($M = 0.10$, $SD = 0.23$), $t(102) = -1.63$, $p = .106$, $d_z = -0.16$, 95% CI $[-0.35, 0.03]$. A linear mixed model of the form $\tau \sim \text{reward} + (1 \mid \text{participant})$ indicated a significant increase of the non-decision time as the reward increased, $\beta = 0.02$, $SE = 0.00$, $t(308.00) = 3.47$, $p < .001$ (see Figure 8d). That is, the non-decision time for the highest reward level ($M = 1.03$, $SD = 0.18$) was higher than for the lowest reward level ($M = 0.99$, $SD = 0.19$), $t(102) = 3.67$, $p < .001$, $d_z = 0.36$, 95% CI $[0.16, 0.56]$. In the congruent group of experiment 2, participants’ median reaction times (in seconds) were slower when they made a decision for a high ($M = 1.90$, $SD = 0.47$) vs. a low reward ($M = 1.84$, $SD = 0.39$), $t(102) = 2.48$, $p = .015$, $d_z = 0.24$, 95% CI $[0.05, 0.44]$.

For the incongruent group, we investigated whether the shown reward had an effect on whether participants responded “old” or “new”. That is, we ran a drift diffusion model on all trials with the outcome variable “old” (lower boundary) or “new” (upper boundary). For the complex model, we let boundary separation, the starting point, the drift rate, and the non-decision time vary by reward level (50, 750, 1450 or 2150 gems). Since no participants had to be excluded due to a lack of trials, the model was run on all $n = 102$ participants. When using the difference in BIC as criterion, the complex model was preferred in 100.00 % of cases. When using the difference in AIC as criterion, the complex model was preferred in 100.00 % of cases. According to a likelihood ratio test between the two models, the complex model provided a significantly better fit in 60.78 % of cases.

To estimate the effect of reward on boundary separation α , we ran a linear mixed model of the form $\alpha \sim \text{reward} + (1 \mid \text{participant})$. There was no significant effect of reward on boundary separation, $\beta = 0.01, SE = 0.01, t(305.00) = 1.54, p = .124$ (see Figure 9a). That is, boundary separation for the highest reward level ($M = 2.12, SD = 0.35$) was not significantly different from boundary separation for the lowest reward level ($M = 2.09, SD = 0.34$), $t(101) = 1.71, p = .090, d_z = 0.17, 95\% CI [-0.03, 0.36]$. To estimate the effect of reward on the starting point β , we ran a linear mixed model of the form $\beta \sim \text{reward} + (\text{reward} \mid \text{participant})$. The starting point decreased as the reward increased, $\beta = -0.02, SE = 0.00, t(101.00) = -3.87, p < .001$ (see Figure 9b). That is, the starting point for the highest reward level ($M = 0.48, SD = 0.09$) was significantly lower (closer to “old”) than the starting point for the lowest reward level ($M = 0.52, SD = 0.08$), $t(101) = -3.76, p < .001, d_z = -0.37, 95\% CI [-0.57, -0.17]$. A linear mixed model of the form $\delta \sim \text{reward} + (\text{reward} \mid \text{participant})$ revealed that the drift rate decreased as the reward increased, $\beta = -0.04, SE = 0.01, t(100.99) = -3.54, p < .001$ (see Figure 9c). That is, the drift rate for the highest reward level ($M = 0.01, SD = 0.26$) was significantly lower than the drift rate for the lowest reward level ($M = 0.11, SD = 0.24$), $t(101) = -3.71, p < .001, d_z = -0.37, 95\% CI [-0.57, -0.17]$. A linear mixed model of the form $\tau \sim \text{reward} + (\text{reward} \mid \text{participant})$ revealed that the non-decision time increased as the reward increased, $\beta = 0.02, SE = 0.01, t(101.00) = 3.13, p = .002$ (see Figure 9d). That is, the non-decision time for the highest reward level ($M = 1.01, SD = 0.26$) was significantly higher than for the lowest reward level ($M = 0.98, SD = 0.26$), $t(101) = 2.82, p = .006, d_z = 0.28, 95\% CI [0.08, 0.48]$. In the incongruent group of experiment 2, participants’ median reaction times (in seconds) are again slower when they make a decision for a high ($M = 1.90, SD = 0.49$) vs. a low reward ($M = 1.84, SD = 0.46$), $t(101) = 3.45, p < .001, d_z = 0.34, 95\% CI [0.14, 0.54]$.”

Results Experiment 3:

“For experiment 3, we investigated whether the reward had an effect on whether participants responded “old” or “new” in the congruent group. That is, we ran a drift diffusion model on all trials with the outcome variable “old” (lower boundary) or “new” (upper boundary). For the complex model,

we let boundary separation, the starting point, the drift rate, and the non-decision time vary by reward level (£ 0.2 or 10). Since no participants had to be excluded due to a lack of trials, the model was run on all $n = 187$ participants. When using the difference in BIC as criterion, the complex model was preferred in 99.47 % of cases. When using the difference in AIC as criterion, the complex model was preferred in 99.47 % of cases. According to a likelihood ratio test between the two models, the complex model provided a significantly better fit in 39.04 % of cases.

Boundary separation for the high reward level ($M = 2.15, SD = 0.35$) did not significantly differ from boundary separation for the low reward level ($M = 2.15, SD = 0.35$), $t(186) = -0.04, p = .970, d_z = 0.00, 95\% CI [-0.15, 0.14]$ (see Figure 11a). The starting point for the high reward level ($M = 0.51, SD = 0.07$) was not significantly different from the starting point for the low reward level ($M = 0.50, SD = 0.06$), $t(186) = -1.18, p = .238, d_z = -0.09, 95\% CI [-0.23, 0.06]$ (see Figure 11b). The drift rate for the high reward level ($M = 0.00, SD = 0.22$) was significantly lower than the drift rate for the low reward level ($M = 0.05, SD = 0.21$), $t(186) = 3.05, p = .003, d_z = 0.22, 95\% CI [0.08, 0.37]$ (see Figure 11c). The non-decision time for the high reward level ($M = 0.83, SD = 0.33$) did not significantly differ from the non-decision time for the low reward level ($M = 0.82, SD = 0.31$), $t(186) = -0.49, p = .625, d_z = -0.04, 95\% CI [-0.18, 0.11]$ (see Figure 11d). In experiment 3, median reaction times do not significantly differ between high ($M = 1.76, SD = 0.40$) and low reward decisions ($M = 1.75, SD = 0.40$), $t(186) = 1.24, p = .216, d_z = 0.09, 95\% CI [-0.05, 0.23]$.

We now discuss these results in the discussion of the main manuscript:

“While our conclusions are based on our preregistered mixed model analyses, it is helpful to look at the data from a different perspective. Our alternative analysis based on drift diffusion models, which take reaction times into account, generally confirms our findings that reward influences participants’ performance at retrieval. In experiment 1, we found that reward expectations do not bias participants in the direction of a hit or false alarm (starting point). Note that the starting point cannot be interpreted as equivalent to a decision criterion in this model, because the outcome was “hit”/“false alarm”, and not “old”/“new”. That is, we cannot draw any conclusions about decision criteria with this model. Likewise, boundary separation (i.e., the amount of evidence collected before making a decision) was not significantly affected by reward. The drift rate increased as reward expectations increased, which means evidence accumulation was faster and more accurate when participants expected a higher reward, potentially due to more noisy memory traces for low rewards. There was a trend towards a decreasing non-decision time as the reward increased; the non-decision time was higher for low rewards than for high rewards. That is, participants needed more time for components that are not part of the decision process (e.g., early

stimulus processing, preparing the motor response) when the reward was very low. Memory processes during early retrieval of stimuli might also contribute to the non-decision time. According to the REM (retrieving effectively from memory⁵¹) model, each item in memory is stored as a vector of features. During a recognition memory test, the probe vector of the test item is compared to the vectors in memory to determine whether the item is old or new. While the comparison of the vectors could be regarded as part of the evidence accumulation phase, earlier phases of the REM model, e.g., compiling an initial set of vectors that the probe vector is compared to, could affect the non-decision time. Within this framework, it might be more effortful to generate a set of possible matches for low-reward items, because their representation in memory is weaker.

For both the congruent and incongruent group of experiment 2, we find that the higher the reward, the more participants are biased towards responding “old”, which corresponds to the effect of reward on the decision criterion we find in the main analysis. Boundary separation, i.e., the amount of evidence needed in order to make a decision, is not affected by reward in both groups. We find a reduced drift rate as the reward increases (a trend for the congruent group; a significant effect for the incongruent group). This may seem at odds with the findings of experiment 1, where the drift rate increased for higher rewards. However – in contrast to experiment 1 – in experiment 2, the reward information is presented before the test stimulus is revealed. Participants might spend additional effort in stimuli labelled as high rewards, even if they do not recognize them. This might make the evidence accumulation for stimuli labelled as high rewards more noisy, resulting in an increased drift rate. In experiment 1, on the other hand, participants might quickly skip over stimuli that do not have a strong representation in memory. A similar explanation could apply to the non-decision time, which also increases as the reward increases: Participants might spend more time on processing a stimulus following a high-reward cue during the test phase, or generating potentially matching vectors in memory.

The findings of the drift diffusion models for experiment 3 follow a different pattern. Reward did not affect the starting point, but the drift rate was significantly lower for the high reward level. This could be interpreted as evidence that processing two reward levels is different from processing four reward levels. The absence of an effect of reward on the non-decision time might explain why we do not find an effect of reward on memory sensitivity in the main analyses. Increased effort for processing the test stimulus, or generating matching vectors in memory, might be required for effective retrieval. The reason for this might be that with two reward categories, there are more stimuli per category – potentially too many to prioritize the high rewards.

Taken together, these findings suggest that participants spend more effort when a stimulus is announced as a high-reward item, regardless of whether its memory representation is strong. In

contrast, decisions based on reward expectations require less effort, because resources are not “wasted” on stimuli that are only represented weakly in memory. Our results might further indicate that non-decision processes might be related to memory sensitivity. Lastly, it should be noted that because they also take reaction times into account, the parameters of drift diffusion models do not directly correspond to signal detection measures such as memory strength and decision bias. Thus, the drift diffusion models reported here provide complementary information to the analyses in the main manuscript, rather than challenging their interpretations.”

2. The authors say that a BIC/AIC difference of >2 is required to support the more complex model. This does not make sense, as the BIC/AIC already account for complexity. Thus, a difference of >0 is sufficient. If the authors distrust these criteria (some argue that they are too lenient), then they need to select other(s).

We now use a difference of > 0 as criterion.

3. My biggest concern is about the decision to fix the NDT across reward levels. This is problematic, as it is known that RT go down with increasing rewards/values (called "Magnitude" or "Overall Value" Effect; see for instance work by Angelo Pirrone). It has been argued that this RT effect could be due to response vigour, which would map onto the NDT parameter in the DDM. In particular in the first experiment, the unexpected direction of the boundary separation effect could look different, if NDT is allowed to differ across reward conditions. How do the results look like when the NDT parameter is not fixed?

Thank you for this valuable feedback – we had no hypotheses regarding the NDT, which is why we initially set it as fixed parameter. However, when adding NDT as free parameter, the results change substantially (e.g., there is no effect of reward on boundary separation anymore), and the model fit of the complex model improves. We have updated the results and discussion accordingly (see above).

Interestingly, our findings are opposite to what would be expected based on the magnitude sensitivity phenomenon. We briefly discuss this:

“In previous literature investigating the influence of reward on decisions and reaction times, the phenomenon of “magnitude sensitivity” has been described: “a choice between equal alternatives of high magnitude is made faster compared with a choice between equal alternatives of low magnitude”, where “magnitude” means the summed value of the alternatives.⁵² In our

experiments, we find the opposite pattern: Participants generally make slower decisions for stimuli preceded by a high reward. The most striking differences between our paradigm and those reported in the literature and magnitude sensitivity is that our participants a) do not choose between multiple alternatives presented on the screen and b) make their decision based on memory. To our knowledge, there are no studies investigating the classical magnitude sensitivity effect in memory-based paradigms. Even though our Motivated Learning Task differs in several aspects from typical magnitude sensitivity paradigms, our data might provide tentative evidence that reward magnitude affects memory-based decisions differently.”

However, you might be interested in a more detailed analysis (not included in the manuscript or the Supplementary Material):

We thought it would be interesting to analyse the data of the incongruent group from the perspective of a magnitude effect by investigating whether the sum of the true and shown reward influences participants’ decisions. However, it appears that performance in the incongruent group is largely driven by the shown reward, not by the sum of both rewards: A mixed model of the form $\text{hit} \ (\theta \text{ or } 1) \sim \text{reward sum} + \text{shown reward} + (1 \mid \text{participant}) + (1 \mid \text{image})$ resulted in a significant effect of shown reward, but no significant effect of reward sum, indicating that the variance in hits is mainly explained by the shown reward, and that the sum of the shown and true reward does not explain any additional variance. This interpretation is supported by a visual representation of how the reward sum, the true and the shown reward relate to the hit rate:

Each panel of the plot shows a different sum of true and shown reward (e.g. a true reward 50 + shown reward 750 = reward sum 800). The x axis shows the true reward, i.e., the shown reward can be inferred from the reward sum. Across the panels, with increasing reward sum, the hit rate increases. However, within each panel, the hit rate decreases as the true reward increases. That is because when the true reward is high, the shown reward must be low, and since the shown reward drives the effect, the hit rate is lower for true rewards within each reward sum panel.

5. Page 62: "..., which results in a stable drift rate". Drift rates are always stable (at least in the 4-parameter DDM). I think what you want to say is "...but they take more time to make a decision when the reward is high, which results in more integration of evidence and thus more accurate decisions."

Since the results of the DDMs have changed, our interpretation and discussion has also changed, the respective sentence has been removed.

5th Feb 24

Dear Ms Nagel,

Your manuscript titled "Memory for Rewards Guides Retrieval" has now been seen by the reviewer who previously had some remaining concerns. Their comments appear below. In light of their advice I am delighted to say that we are happy, in principle, to publish a suitably revised version in Communications Psychology under the open access CC BY license (Creative Commons Attribution v4.0 International License).

We therefore invite you to revise your paper one last time to address the editorial requests. At the same time we ask that you edit your manuscript to comply with our format requirements and to maximise the accessibility and therefore the impact of your work.

EDITORIAL REQUESTS:

SUBMISSION INFORMATION:

OPEN ACCESS:

Communications Psychology is a fully open access journal. Articles are made freely accessible on publication under a CC BY license (Creative Commons Attribution 4.0 International License). This license allows maximum dissemination and re-use of open access materials and is preferred by many research funding bodies.

For further information about article processing charges, open access funding, and advice and support from Nature Research, please visit <https://www.nature.com/commspsychol/article-processing-charges>

At acceptance, you will be provided with instructions for completing this CC BY license on behalf of all authors. This grants us the necessary permissions to publish your paper. Additionally, you will be asked to declare that all required third party permissions have been obtained, and to provide billing information in order to pay the article-processing charge (APC).

* TRANSPARENT PEER REVIEW: Communications Psychology uses a transparent peer review system. On author request, confidential information and data can be removed from the published reviewer

reports and rebuttal letters prior to publication. If you are concerned about the release of confidential data, please let us know specifically what information you would like to have removed. Please note that we cannot incorporate redactions for any other reasons.

* CODE AVAILABILITY: All Communications Psychology manuscripts must include a section titled "Code Availability" at the end of the methods section. We require that the custom analysis code supporting your conclusions is made available in a publicly accessible repository at this stage; please choose a repository that generates a digital object identifier (DOI) for the code; the link to the repository and the DOI must be included in the Code Availability statement. Publication as Supplementary Information will not suffice.

* DATA AVAILABILITY:

[link redacted]

Best wishes,

Marike

Marike Schiffer, PhD
Chief Editor
Communications Psychology

REVIEWERS' COMMENTS:

Reviewer #3 (Remarks to the Author):

I am very impressed by the amount of effort that the authors invested to address the remaining points I had on the previous submission. I did not anticipate that my questions would require this level of revision, but I hope the authors agree that the additional DDM analyses provide an

interesting new perspective to their work.

I have no more comments and (highly) recommend the paper for publication in its current form.

Signed,
Sebastian Gluth